# 3D cell neighbour dynamics in growing pseudostratified epithelia

**Harold Fernando Gómez[1,2], Mathilde Sabine Dumond[1,2], Leonie Hodel[1], Roman Vetter[1,2], Dagmar Iber[1,2]\***

[1]Department of Biosystems Science and Engineering (D-BSSE), ETH Zürich, Basel, Switzerland; [2]Swiss Institute of Bioinformatics (SIB), Basel, Switzerland

**Abstract** During morphogenesis, epithelial sheets remodel into complex geometries. How cells dynamically organise their contact with neighbouring cells in these tightly packed tissues is poorly understood. We have used light-sheet microscopy of growing mouse embryonic lung explants, three-dimensional cell segmentation, and physical theory to unravel the principles behind 3D cell organisation in growing pseudostratified epithelia. We find that cells have highly irregular 3D shapes and exhibit numerous neighbour intercalations along the apical-basal axis as well as over time. Despite the fluidic nature, the cell packing configurations follow fundamental relationships previously described for apical epithelial layers, that is, Euler's polyhedron formula, Lewis' law, and Aboav-Weaire's law, at all times and across the entire tissue thickness. This arrangement minimises the lateral cell-cell surface energy for a given cross-sectional area variability, generated primarily by the distribution and movement of nuclei. We conclude that the complex 3D cell organisation in growing epithelia emerges from simple physical principles.

**\*For correspondence:** dagmar.iber@bsse.ethz.ch

**Competing interests:** The authors declare that no competing interests exist.

## Introduction

Common to all animals and plants, epithelia are a fundamental tissue type whose expansion, budding, branching, and folding is key to the morphogenesis of organs and body cavities. Characterised by apical-basal polarity (*Figure 1a*), epithelial cells adhere tightly to their apical neighbours in a virtually impermeable adhesion belt, form lateral cell-cell junction complexes along the apico-basal axis to provide mechanical stabilisation, and bind tightly to the basal lamina and extracellular matrix (ECM) on the basal side (*Drubin and Nelson, 1996*; *Rodriguez-Boulan and Macara, 2014*; *Shin and Margolis, 2006*). How cell neighbour relationships are organised in these tightly adherent layers, and how these change during tissue and concomitant cell shape changes is poorly understood, despite their importance for cell-cell signalling and the fluidity of the tissue.

Cell neighbour relationships can be most easily studied on epithelial surfaces, and the polygonal arrangements of apical surfaces (*Figure 1b*) have been meticulously analysed (*Classen et al., 2005*; *Escudero et al., 2011*; *Etournay et al., 2015*; *Farhadifar et al., 2007*; *Gibson et al., 2006*; *Gómez-Gálvez et al., 2018*; *Heller et al., 2016*; *Kokic et al., 2019*; *Ramanathan et al., 2019*; *Sánchez-Gutiérrez et al., 2016*). Widely considered to be a reliable proxy for three-dimensional (3D) cell shape, 3D epithelial cell shapes are often depicted as prisms with polygonal faces that retain the same neighbour relationships along the entire apico-basal axis (*Figure 1c*). Cells in curved epithelia are pictured as frustra, which have the same number of sides, but different apical and basal areas. If the curvature differs substantially along the principal axes, as is the case in epithelial tubes, neighbour relationships must change along the apical-basal axis. Prismatoids accommodate the neighbour change at the surface, while scutoids undergo the neighbour change somewhere along the apical-basal axis (*Figure 1c,d*; *Gómez-Gálvez et al., 2018*). However, even though the curvature is the same in both principal directions of spherically shaped epithelia, the neighbour relationships still

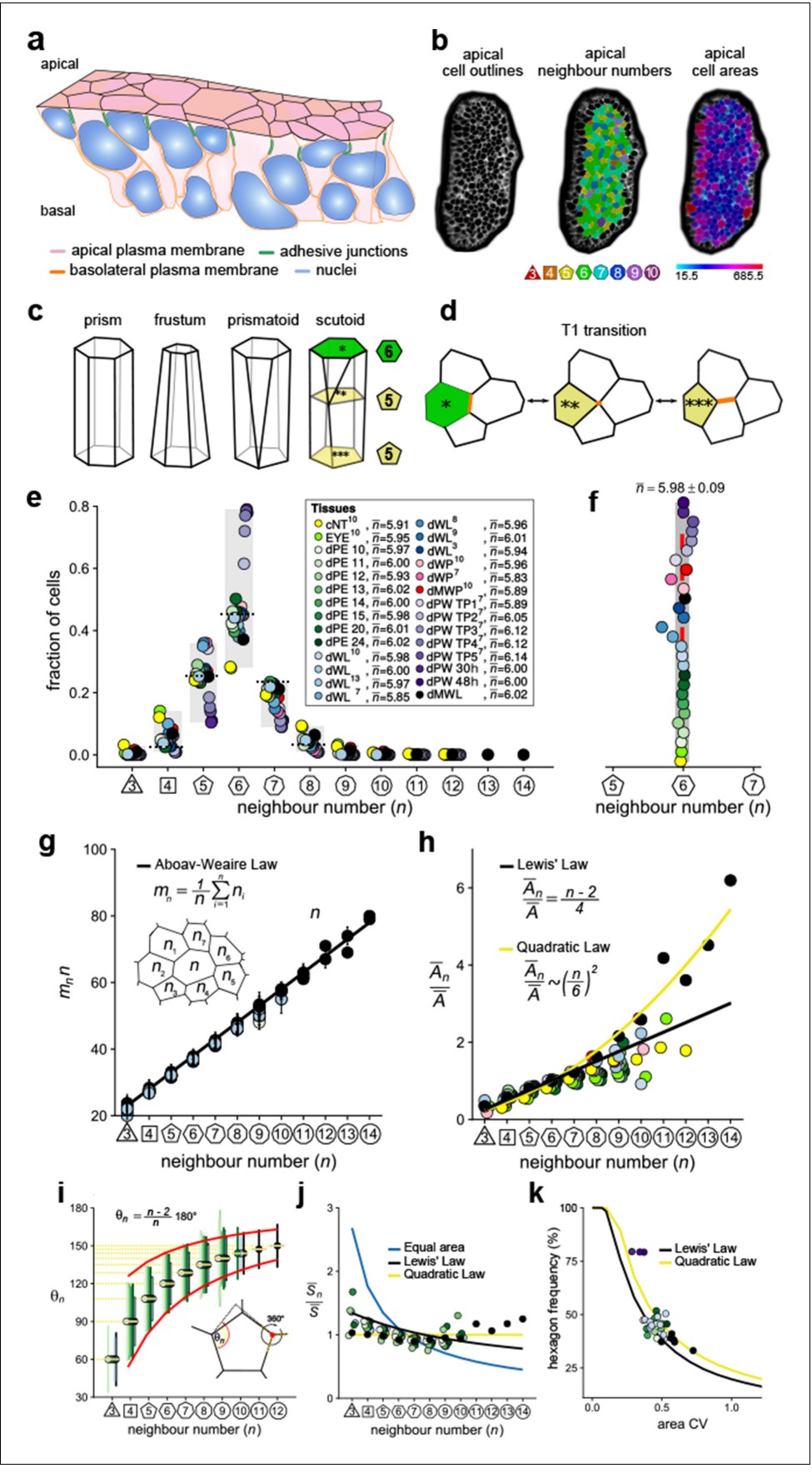

**Figure 1.** Principles of epithelial organisation. (**a**) Schematic representation of an epithelial tissue layer. The cells are polarised between an apical and a basal side. Near the apical side, cells adhere tightly via adhesion junctions (green). Nuclei are depicted in blue. (**b**) Apical surface projection of an embryonic lung bud at E12.5 imaged using light-sheet microscopy. Cell contour segmentations (left) coloured according to neighbour relationships (middle) and area quantifications (right). (**c**) Current shape representations of 3D epithelial cells: prism, frustum, prismatoid, and scutoid. (**d**) Planar cell neighbour exchange (T1 transition). (**e**) Tissues differ widely in the frequency of neighbour numbers. The legend provides the measured average number of cell neighbours for each tissue and the references to the primary data (*Classen et al., 2005*; *Escudero et al., 2011*; *Etournay et al., 2015*; *Farhadifar et al., 2007*; *Gibson et al., 2006*; *Heller et al., 2016*; *Sánchez-Gutiérrez et al., 2016*). Data points for n < 3 were removed as they must present segmentation artefacts. (**f**) The measured average number of cell neighbours is close to the topological requirement ($\bar{n} = 6$) in all tissues; see panel e for the colour code. (**g**) Epithelial tissues follow the AW law (black line). The AW law formulates a relationship between the average number of neighbours, $n$, that a cell has and that its direct neighbours have, $m_n$. The product $m_n \cdot n$ can be determined by summing over all $n_i$. (**h**) The relative average apical cell area, $\bar{A}_n/\bar{A}$, increases with the number of neighbours, $n$, and mostly follows the linear Lewis' law (*Equation 2*, black line), or the quadratic relationship (*Equation 3*, yellow line) in case of higher apical area variability. (**i**) The average internal angle by polygon type is close to that of a regular polygon, $\theta_n = (n-2)/n \cdot 180°$ (yellow lines). To form a contiguous lattice, the angles at each tricellular junction must add to 360°, and the resulting observed deviation in the angles follows the prediction (red line). (**j**) The average normalised side length by polygon type. (**k**) Observed fraction of hexagons versus area coefficient of variation (CV). The curves mark theoretical predictions when polygonal cell layers follow either the linear Lewis' law (*Equation 2*, black line) or the quadratic law (*Equation 3*, yellow line). The colour code in panels g-k is as in panel e, but data is available only for a subset of tissues. The abbreviations in panel (**e**) are as follows: cNT refers to the Chick neural tube epithelium, EYE to the *Drosophila* eye disc, dPE to the *Drosophila* peripodal membrane from the larval eye disc, dWL to the *Drosophila* larval wing disc, dWP to the *Drosophila* pre-pupal wing disc, dMWP to the *Drosophila* mutant wing pre-pupa with reduced expression of myosin II, dPW to the *Drosophila* pupal wing disc, and dMWL to the wing disc with gigas RNAi clones. TPx indicates subsequent but not further specified pupal time points.

differ between the apical and basal sides (*Gómez-Gálvez et al., 2018*), suggesting that effects other than curvature must determine the 3D neighbour arrangements of cells in epithelia.

Given the challenges in visualising 3D neighbour arrangements, most studies to date have focused on apical cell arrangements, and have revealed striking regularities. First, even though the frequencies of neighbour numbers differ widely between epithelial tissues (*Figure 1e*), cells have on average (close to) six neighbours (*Classen et al., 2005*; *Escudero et al., 2011*; *Etournay et al., 2015*; *Farhadifar et al., 2007*; *Gibson et al., 2006*; *Heller et al., 2016*; *Kokic et al., 2019*; *Ramanathan et al., 2019*; *Sánchez-Gutiérrez et al., 2016*; *Figure 1f*). This can be explained with topological constraints in contiguous polygonal lattices, as expressed by Euler's polyhedron formula (*Gibson et al., 2006*; *Rivier and Lissowski, 1982*). Thus, if three cells meet at each vertex, the average number of neighbours in infinitely large contiguous polygonal lattices is exactly

$$\bar{n} = 6. \tag{1}$$

While the average number of neighbours in the entire lattice is (close to) six, the local averages deviate from six, and instead rather closely follow a phenomenological relationship, termed Aboav-Weaire's law (*Aboav, 1970*). According to Aboav-Weaire's law (*Figure 1g*), the average number of neighbours of all $n$ cells that border a cell with $n$ neighbours follows as

$$m(n) = 5 + \frac{8}{n}. \tag{2}$$

Finally, the average apical area, $\bar{A}_n$, of cells with $n$ neighbours is linearly related to the number of cell neighbours, $n$ (*Figure 1h*, black line), a relation termed Lewis' law (*Lewis, 1928*),

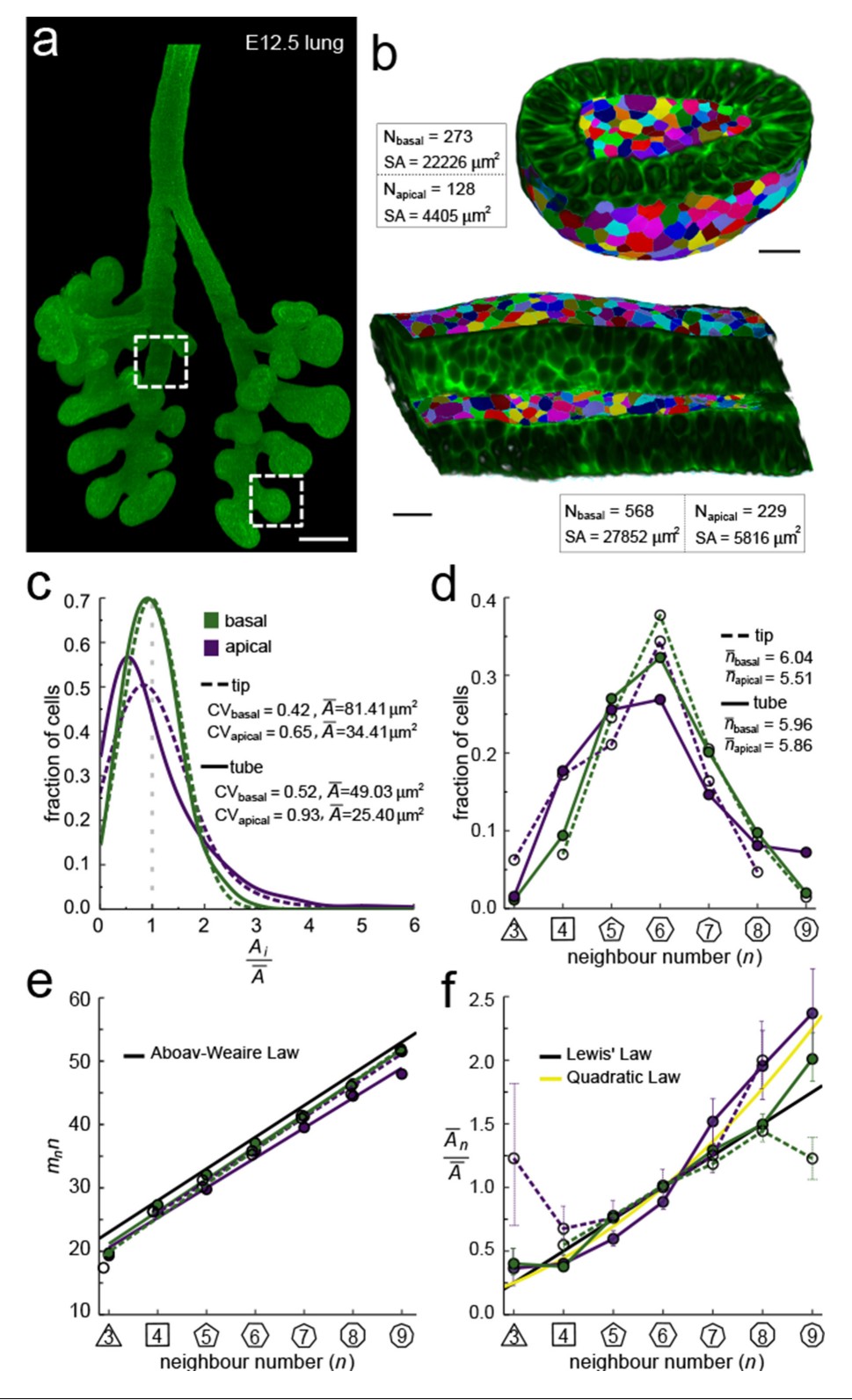

**Figure 2.** Apical and basal epithelial organisation. (a) Epithelium of E12.5 Shh[GC/+]; ROSA[mT/mG] mouse embryonic lung imaged using light-sheet microscopy. Scale bar 200 μm. Corresponding 2D sections are shown in *Figure 2—figure supplement 1*, and *Figure 2—video 1*. (b) Apical and basal 2.5D cell segmentation overlays on imaged tip and tube sections (dotted boxes in panel a). An illustration of the 2.5D segmentation workflow is presented in *Figure 2 continued on next page*

*Figure 2 continued*

*Figure 2—figure supplement 2*. Number of cells (N) and segmented surface areas (SA) are given. Cells are coloured using random labels. Scale bars 20 µm. (c) Normalised apical and basal cell area distributions in the tip (broken lines) and tube (solid lines) datasets. The colour code in panel c is reused in panels d-f. (d) Frequencies of neighbour numbers on the apical and basal sides in the tip and tube datasets. (e) The apical and basal layers follow the AW law (black line). SEM is smaller than symbols. (f) The normalised average cell area, $\frac{\bar{A}_n}{\bar{A}}$, increases with the number of neighbours, $n$. The basal cells (green) follow Lewis' law (*Equation 2*, black line), while the apical cells (purple) follow the quadratic relationship (*Equation 3*, yellow line). SEM as error bars.

The online version of this article includes the following video, source data, and figure supplement(s) for figure 2:

**Source data 1.** 2.5D neighbour number and area data.
**Source data 2.** 2.5D AW data.
**Figure supplement 1.** Embryonic mouse lung rudiments.
**Figure supplement 2.** Workflow for surface cell segmentations.
**Figure supplement 3.** CUBIC clearing of embryonic tissue.
**Figure 2—video 1.** Optically cleared mouse lung rudiment image stacks.
https://elifesciences.org/articles/68135#fig2video1

$$\frac{\bar{A}_n}{\bar{A}} = \frac{(n-2)}{4}. \tag{3}$$

Here, $\bar{A}$ refers to the average apical cell area in the tissue.

We have recently shown that Aboav-Weaire's law and Lewis' law are a direct consequence of a minimisation of the lateral cell-cell contact surface energy (*Kokic et al., 2019*; *Vetter et al., 2019*). The lowest lateral cell-cell contact surface energy is obtained in a regular polygonal lattice because regular polygons have the smallest perimeter per polygonal area. The distribution of apical cell sizes that emerges from cell growth and division is, however, such that epithelial tissues cannot organise into perfectly regular polygonal lattices. By adhering to Aboav-Weaire's law and Lewis' law, cells assume the most regular lattice. In particular, by following Aboav-Weaire's law, the internal angles are closest to that of a regular polygon, while adding up to 360° at each tricellular junction (*Figure 1i*; *Vetter et al., 2019*). And by following the relationship between polygon area and polygon type as stipulated by Lewis' law (*Equation 2*), the difference in side lengths, $\bar{S}_n/\bar{S}$, is minimised between cells (*Figure 1j*; *Kokic et al., 2019*). The side lengths would be equal (*Figure 1j*, yellow line), if cells followed a quadratic relation of the form

$$\frac{A_n}{\bar{A}} = \frac{n}{6} \cdot \frac{\tan\left(\frac{\pi}{6}\right)}{\tan\left(\frac{\pi}{n}\right)} \sim \left(\frac{n}{6}\right)^2. \tag{4}$$

This quadratic relation (*Figure 1h*, yellow line), however, requires a larger area variability than is observed in most epithelia imaged to date. Accordingly, the predicted quadratic relation had not been previously reported, but could be confirmed experimentally by us by increasing the apical area variability (*Kokic et al., 2019*).

Given the relationship between apical area and neighbour numbers as stipulated by *Equations 1,3,4*, the apical area variability emerges as the key determinant of apical epithelial organisation, and the theory correctly predicts how the fraction of hexagons in the tissue depends on the apical area variability, as can be quantified by the coefficient of variation (CV = std/mean) (*Figure 1k*; *Kokic et al., 2019*). As such, active processes such as growth, cell division, cell death and extrusion, cell intercalation and apical constriction determine the variability of the apical areas and thus determine apical organisation indirectly. Taken together, the apical organisation of epithelia can be understood based on the principles of lateral cell-cell contact surface energy minimisation.

In this work, we leverage these theoretical insights along with light-sheet fluorescence microscopy to study 3D epithelial organisation, both in cleared and growing pseudostratified epithelia. We find that cells have complex 3D shapes with numerous neighbour transitions along their apical-basal axis as well as over time. We show that much as on the apical side, the variation of the cross-sectional areas along the apical-basal axis defines the epithelial organisation at all times and across the entire tissue thickness. The observed neighbour arrangement minimises the lateral cell-cell surface energy

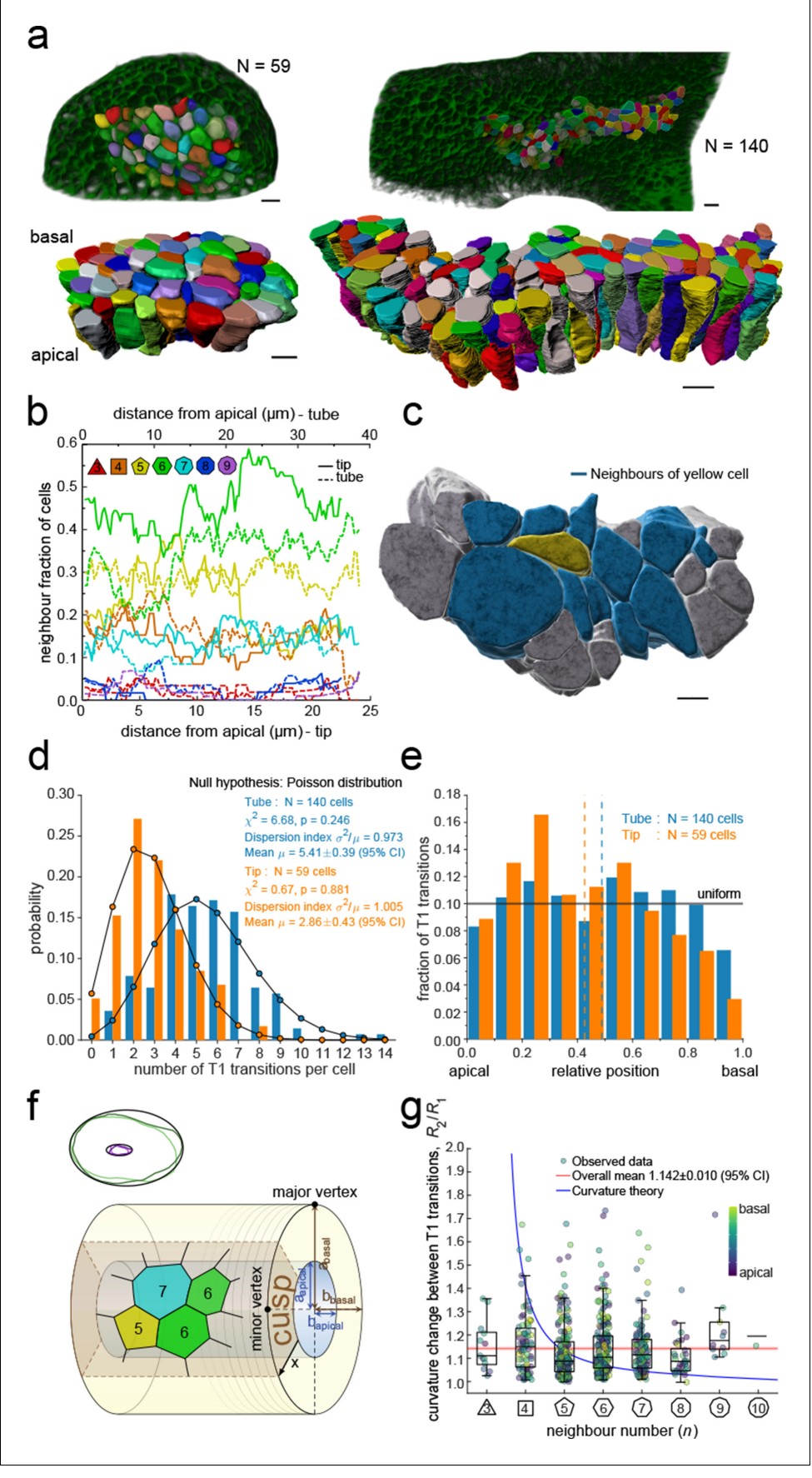

**Figure 3.** 3D epithelial organisation. (a) 3D iso-surfaces of segmented epithelial tip (N=59) and tube (N=140) cells from a Shh$^{GC/+}$; ROSA$^{mT/mG}$ E12.5 mouse lung rudiment imaged using light-sheet microscopy. Morphometric quantifications of cell boundary segmentations along the apical-basal axis were used to study spatial T1 transitions. The 3D segmentation workflow is introduced in *Figure 3—figure supplement 1* and *Figure 3—videos 1–3* illustrate the rendered epithelial tip, tube and all segmented volumes. Scale bars 10 µm. (b) Frequency of neighbour numbers as quantified along the apical basal axis in the tip (solid lines) and tube (broken lines) datasets. (c) Extent of neighbour contacts (center cell in yellow and neighbours in blue) in 3D as viewed from the apical side. Scale bars 5 µm. (d) Probability distributions of the lateral T1 transitions for tip (total=169, mean=2.86, N=59) and trunk (total=746, mean=5.41, N=140) datasets are consistent with Poisson distributions. (e) Normalised apical-basal distribution of T1 transitions for all cells shows no apical-basal bias, except for fewer transitions close to the basal surface. (f). Schematic of a tubular epithelium with elliptic cross section. The analysed cells are located in the cusp (brown) of the tube, where the local tissue curvature is close to that of the minor ellipse vertex. Subpanel illustrates ellipse fitting to apical (purple) and basal (green) domain boundaries. Lighter outlines correspond to the most proximal segment of the tube in *Figure 2b*, while darker ones to the most distal section. (g) The predicted impact of a curvature effect on T1 transitions decreases with increasing cell neighbour numbers (blue line). The measured T1 transitions for different neighbour numbers do not support a curvature effect (dots, boxplots, and red line). Boxplots indicate the median, 25% and 75% percentiles of the data.

The online version of this article includes the following video, source data, and figure supplement(s) for figure 3:

**Source data 1.** 3D neighbour number and area data.
**Source data 2.** 3D curvature change data.
**Figure supplement 1.** Workflow for 3D epithelial cell and nuclear segmentations.
**Figure 3—video 1.** 3D contour surfaces.
https://elifesciences.org/articles/68135#fig3video1
**Figure 3—video 2.** 3D nuclear and tube cell segmentations.
https://elifesciences.org/articles/68135#fig3video2
**Figure 3—video 3.** 3D tip cell segmentations.
https://elifesciences.org/articles/68135#fig3video3

---

for a given cross-sectional area variability. The cross-sectional areas vary as a result of active cell processes, most prominently including interkinetic nuclear migration (IKNM). We conclude that the complex 3D cell organisation in growing epithelia emerges from simple physical principles.

## Results

### Apical and basal epithelial organisation

We started by exploring the apical and basal cellular organisation in epithelial tubes and buds (*Figure 2a*). To this end, we imaged CUBIC-cleared mouse embryonic (E12.5) lung rudiments from a Shh$^{GC/+}$; ROSA$^{mT/mG}$ background using light-sheet microscopy, and segmented the fluorescent membrane boundaries of over 400 cells per dataset in 2.5D (*Figure 2b*, *Figure 2—figure supplement 2*). The apical and basal surfaces are both curved and thus differ in their total areas, that is, the total segmented apical area is about 5-fold smaller than the basal area (*Figure 2b*). We detected less than half as many cells on the apical side, and the mean cross-sectional cell area of apical cells is therefore on average only twofold smaller than that of basal cells, while the area variability, measured as area CV, is higher (*Figure 2c*). Notably, the frequencies of the different neighbour numbers are not identical on the apical and basal side (*Figure 2d*), suggesting that the neighbour relationships change along the apical-basal axis, both in the tube and tip datasets. This observation is consistent with previous reports (*Gómez-Gálvez et al., 2018*; *Ramanathan et al., 2019*; *Rupprecht et al., 2017*). The change in neighbour relationships has previously been attributed to a curvature effect in tubes, but the neighbour changes in spherical geometries cannot be explained with curvature alone, since prismatic cells fully accommodate the cell dilation from equal curvature changes in both directions (*Gómez-Gálvez et al., 2018*; *Rupprecht et al., 2017*), suggesting that mainly other effects determine epithelial organisation.

So how can we explain the difference in apical and basal epithelial organisation in both datasets? We have previously shown that the apical organisation emerges from the minimisation of the overall lateral cell-cell contact surface energy (*Kokic et al., 2019*; *Vetter et al., 2019*). Aboav-Weaire's law (*Equation 2*, *Figure 1g*) and the linear Lewis' law or the quadratic relationship (*Equations 3,4*,

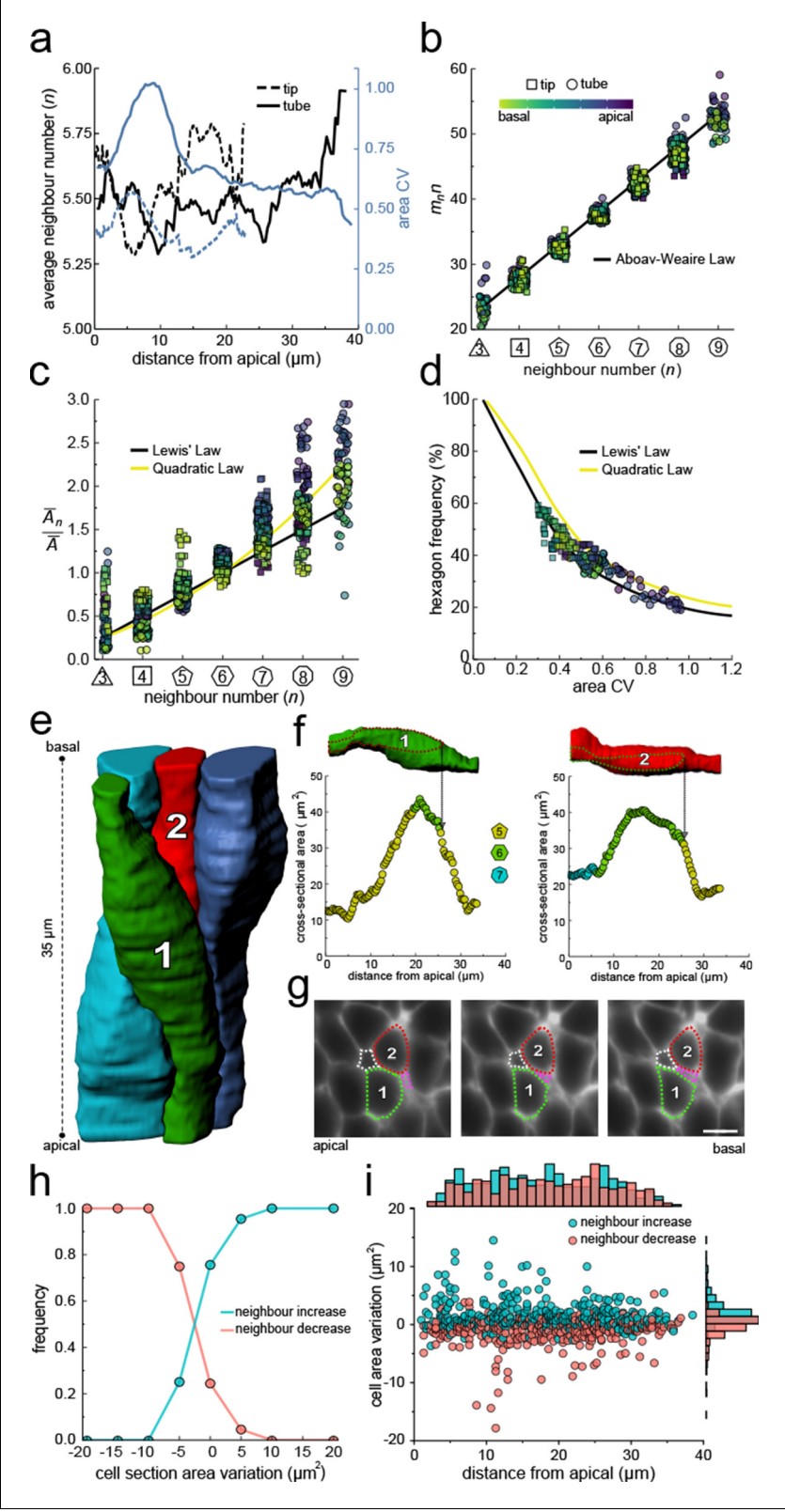

**Figure 4.** Neighbour changes along the apical-basal axis are driven by changes in cross-sectional area. (a) Average number of neighbours (black) and area CV (blue) along the apical basal axis in the tip and tube datasets. (b) All epithelial layers follow the AW law (black line). The colour code in panels c, d follow that in panel b. (c) All epithelial layers follow Lewis' law (*Equation 2*, black line) in case of low, and the quadratic relationship

*Figure 4 continued on next page*

*Figure 4 continued*

(*Equation 3*, yellow line) in case of high cell area variability. (**d**) Observed fraction of hexagons versus area CV for segmented cell layers along the apical-basal axis. The lines mark the theoretical prediction if polygonal cell layers follow either the linear Lewis' law (black line) or the quadratic law (yellow line). (**e**) 3D iso-surfaces of four segmented epithelial cells in a CUBIC-cleared Shh$^{GC/+}$; ROSA$^{mT/mG}$ E12.5 distal lung tube, with 140 3D segmented epithelial cells (*Figure 3—figure supplement 1c*). (**f**) Cross-sectional area and cell neighbour number along the apical-basal axis for marked cells in panel e. Dotted lines indicate contact with one another. (**g**) Lateral cross-sections illustrating a T1 transition along the apical-basal axis (0.664 µm in-between frames). Scale bar 6 µm. (**h**) An increase in the cell cross-sectional area increases the frequency of a neighbour number increasing spatial T1 transitions, and vice versa. (**i**) Apical-basal distribution of spatial T1 transitions according to neighbour increase or decrease and cross-sectional area variation.

The online version of this article includes the following source data for figure 4:

**Source data 1.** 3D area CV and avg. neighbour number data.

**Source data 2.** 3D AW data.

**Source data 3.** Neighbour exchange/type data.

*Figure 1h*) emerge as global organisation laws from this physical constraint, and ensure that the angles are closest to that of a regular polygon (*Figure 1i*, yellow lines), and that the side lengths are the most equal (*Figure 1j*, yellow line). We now find that these hold not only for the apical, but also for the basal datasets (*Figure 2e,f*). Consistent with our theory, the apical layers, which have a larger area variability than the basal layers (*Figure 2c*), follow the quadratic law (yellow line) rather than the linear Lewis' law (black line).

We conclude that basal layers follow the same organisational principles as apical layers, such that their organisation can also be explained with a minimisation of the lateral cell-cell contact surface energy. Accordingly, the observed difference in overall neighbour relationships (*Figure 2e,f*) is a consequence of the difference in the cross-sectional area distributions (*Figure 2c*). So, why do the normalised area distributions differ between the apical and basal sides in both the tube segment and the bud, and how do they change along the apical-basal axis?

## 3D organisation of epithelia

To explore the physical principles behind 3D epithelial cell organisation, we 3D segmented 140 cells from a tube segment and 59 cells from a bud segment in CUBIC-cleared, light-sheet imaged embryonic lung explants (*Figure 3a*, *Figure 2—figure supplement 1*, *Figure 2—video 1*, *Figure 3—videos 1–3*). By interpolating between equally spaced sequential contour surfaces (every 1.66 µm in the tube and every 1.72 µm in the bud dataset) along the apical-basal axis, accurate volumetric reconstructions of cell morphology were obtained that allowed for the extraction of morphometric quantifications along the apical-basal axis. In both datasets, the 3D organisation of epithelial cells is highly complex, and cell neighbour relationships change continuously along the apical-basal axis (*Figure 3b*). As a result, cells are in direct physical contact not only with the cells that are neighbours on the apical side, but also with cells that appear two or even three cell diameters apart (*Figure 3c*).

Remarkably, we record up to 14 cell neighbour changes per cell in the tube and up to eight in the tip, between adjacent cross-sections along the apical-basal cell axis (*Figure 3d*). We will refer to these neighbour changes as lateral T1 transitions, or T1L. The mean relative apical-basal position for the lateral T1 transitions is 0.489±0.020 (95% CI), and there is no clear apical or basal tendency, though fewer transitions are observed close to the basal surface (*Figure 3e*). The dispersion index, that is, the ratio of the variance $\sigma^2$ and the mean number µ of transitions per cell, which equals unity for a Poisson distribution, is close to unity for both samples (*Figure 3d*). The chi-squared test also confirms that the number of apical-basal T1 transitions per cell is Poisson-distributed (*Figure 3d*). A Poisson distribution models the probability of a number of independent random events occurring in a given interval at a constant average rate. The consistency with a Poisson distribution, therefore, suggests a stochastic basis to the 3D organisation of epithelial cells.

The large number of observed T1L transitions and their distribution along the apical-basal axis challenges the recently popularised notion of curvature-driven scutoids as cell building blocks for epithelia (*Gómez-Gálvez et al., 2018*). To further examine the potential influence of tissue curvature

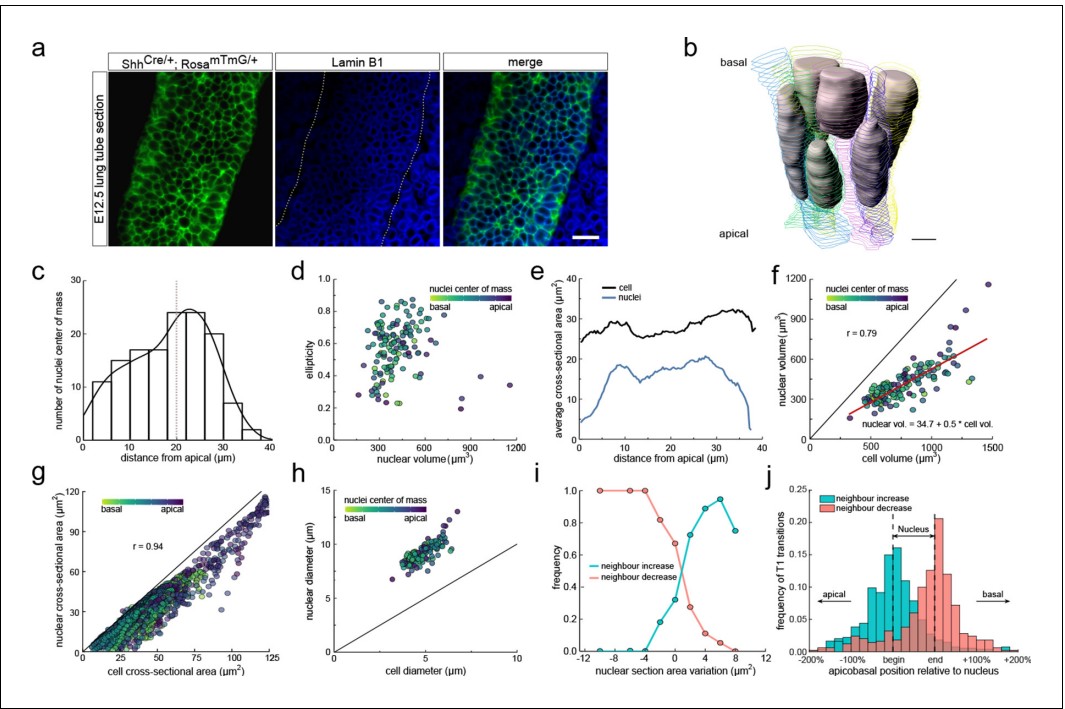

**Figure 5.** Changes in cross-sectional area as a result of interkinetic nuclear migration (IKNM). (**a**) Light-sheet microscopy longitudinal sections of an E12.5 CUBIC-cleared lung tube carrying the Shh^GC/+; ROSA^mT/mG reporter allele (green epithelium) and immunostained for lamin B1 (blue nuclear envelopes). Morphometric quantifications of 3D iso-surfaces (N=140) and cell segmentations along the apical-basal axis were used to study the nature of cross-sectional area variation and the effect of IKNM (**Figure 3—figure supplement 1**). Scale bar 20 μm. (**b**) Sequential cell membrane contour surfaces and nuclear iso-surfaces for six epithelial cells. By interpolating between contours and creating iso-surfaces, 3D shapes can be accurately extracted (**Figure 3—video 2** and **Figure 5—video 1**). Scale bar 7 μm. (**c**) Distribution of nuclei center of mass along the apical-basal axis. (**d**) Nuclear ellipticity and volume distributions along the apical-basal axis. (**e**) Average cross-sectional area distribution along the apical-basal axis for all cells (black) and nuclei (blue). (**f**) Nuclear and cellular volumes of 140 segmented cells are correlated (r=0.79). Line of best fit in red. (**g**) The cell and corresponding nuclear cross-sectional areas along the apical-basal axis are highly correlated (r=0.94). (**h**) Diameters of nuclei and cells based on the measured nuclear and cell volumes if nuclei were perfect spheres, and cells were perfect cylinders of the measured height. Given the larger nuclear diameter, nuclei must deform in order to fit within cells. (**i**) An increase in the nuclear cross-sectional area increases the frequency of a neighbour-number-increasing spatial T1 transitions, and vice versa. (**j**) The largest number of changes in neighbour relationships occur at the apical and basal limits of the nucleus for all cells, where cross-sectional areas change sharply.

The online version of this article includes the following video, source data, and figure supplement(s) for figure 5:

**Source data 1.** 3D neighbour number, cell area, and nuclear area data.
**Source data 2.** Volume and ellipticity data.
**Source data 3.** 3D cell and nuclear diameter data.
**Figure supplement 1.** Light-sheet imaging of a stained embryonic mouse lung rudiment.
**Figure 5—video 1.** 3D nuclear iso-surface segmentations and cell membrane contours.
https://elifesciences.org/articles/68135#fig5video1

on T1L transitions, we measured the apical-basal distance between two consecutive neighbour number changes for each cell in the tube dataset and recorded at which local tissue curvature they occur. For this analysis, we excluded ce ll portions from the apical end to the first transition and from the last transition to the basal end to reduce boundary effects, that is, only interior segments between transitions were considered. The mean apical-basal distance between two transitions is 17.89 ± 0.66 μm (95% CI). Local tissue curvature was approximated by fitting ellipses to the apical and basal surface boundaries of the tubular epithelium in 624 equidistant sections perpendicular to the main tube axis. (**Figure 3f**). Epithelial tubes in the developing lung are often collapsed (**Conrad et al., 2021**),

making their apical and basal surfaces nearly elliptic in shape. The semi-axes of the fitted ellipses were then averaged over all sections to obtain the semi-axes $a_{\text{apical}}$, $a_{\text{basal}}$, $b_{\text{apical}}$, $b_{\text{basal}}$. Since our sample of 140 cells was segmented from a region close to the cusp of the nearly elliptical tube, a reasonably close estimate of the local tissue curvature where a T1L transition occurs is given by a linear interpolation between the curvature at the minor vertices of the apical and basal ellipses, according to the relative apical-basal position of the transition. The minor curvature of an ellipse with major and minor semi-axes $a$ and $b$ is given by $b/a^2$. Therefore, we estimate the local radius of curvature $R$ by

$$R(x) = \frac{\left(a_{\text{apical}} + x\left(a_{\text{basal}} - a_{\text{apical}}\right)\right)^2}{b_{\text{apical}} + x\left(b_{\text{basal}} - b_{\text{apical}}\right)}$$

where $x \in [0,1]$ is the relative apical-basal location of the T1L transition. The examined tissue exhibits an average curvature fold change of $R(1)/R(0) = 2.21$ from the basal to the apical side. Denoting by $R_1$ and $R_2$ the radii of curvature between two adjacent T1 transitions along the apical-basal axis of a cell, we find that the distribution of curvature fold change $R_2/R_1$ shows no significant dependency on the number of neighbours $n$ the cell has along that portion of the cell (*Figure 3g*). The mean curvature fold change per apical-basal T1L transition per cell is $<R_2/R_1> = 1.142 \pm 0.010$ (95% CI). By extending the theory of scutoids (*Gómez-Gálvez et al., 2018*) to multiple T1L transitions per cell, we have derived a quantitative estimate of how tissue curvature would translate into the number of neighbour exchanges within that framework (Supplementary Material). If curvature changes were a main driver of T1L transitions, cells with smaller neighbour numbers n would be expected to change n over a much larger curvature fold change than cells with many neighbours (*Figure 3g*, blue line). However, we observe no systematic dependency of the curvature fold change on the number of neighbours the cell has along that portion of the cell in the developing mouse lung epithelium (*Figure 3g*). From this, we conclude that tissue curvature affects cell neighbourhood rearrangements through the tissue thickness at most mildly.

## Neighbour changes along the apical-basal axis are driven by changes in cross-sectional area variation

Other than curvature effects, what else could drive the observed changes in neighbour relationships along the apical-basal axis? We notice that much as the apical and basal layers, each layer along the apical-basal axis behaves according to the three relationships previously described for the apical side, that is, Euler's polyhedron formula (*Equation 1*, *Figure 4a*), Aboav-Weaire's law (*Equation 2*, *Figure 4b*), and Lewis' law (*Equations 3, 4*, *Figure 4c*). As predicted by the theory based on the minimisation of the lateral cell-cell energy (*Kokic et al., 2019*), the layers with a large area variability (*Figure 4a*) follow the quadratic law (yellow line) and those with a lower area variability the linear Lewis' law (black line). The fraction of hexagons also follows the predicted relationship with the cross-sectional area variability (*Figure 4d*). We conclude that the neighbour relationships of epithelial cells along the entire apical-basal axis can be explained with a minimisation of the lateral cell-cell contact surface energy, as previously revealed for the apical layer.

If epithelial cell neighbour relationships are indeed driven by a minimisation of the total lateral cell-cell contact surface energy then the T1L transitions along the apical-basal axes should be driven by changes in the cross-sectional area along the apical-basal axis. If we analyse four 3D segmented cells (*Figure 4e*) in detail, we indeed see how an increase in the cross-sectional area results in an increase in the neighbour number, and vice versa (*Figure 4f*) via lateral T1 transitions (*Figure 4g*). As the cell neighbour arrangements represent global minima, the local analysis does, of course, not provide a perfect correlation. When we consider all 140 segmented cells in the tube segment with their 746 cell neighbour exchanges between adjacent cross-sections (*Figure 3e*), then we find that the frequency of T1L transitions along the apical-basal axis is indeed higher, the larger the increase in cross-sectional area, and vice versa (*Figure 4h,i*).

## Changes in cross-sectional area as a result of interkinetic nuclear migration (IKNM)

So, what determines the cross-sectional cell areas in each layer? In pseudostratified epithelia, mitosis is restricted to the apical surface (*Gundersen and Worman, 2013*). Depending on the average

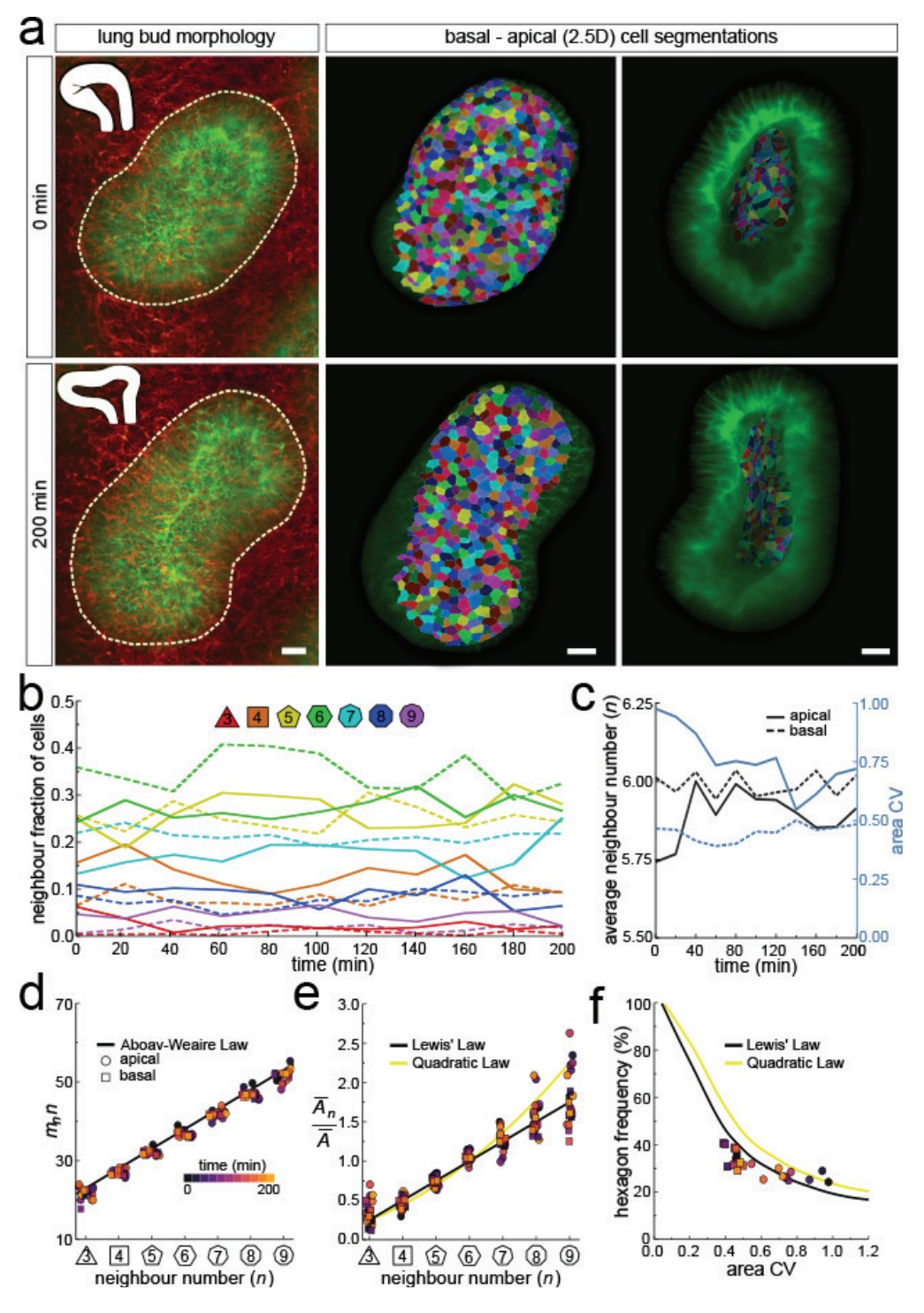

**Figure 6.** Dynamics of apical and basal epithelial organisation. (**a**) Timelapse light-sheet microscopy series of a cultured mouse E12.5 distal lung bud expressing the Shh$^{GC/+}$; ROSA$^{mT/mG}$ reporter (green epithelium, and red mesenchyme), imaged every 20 min (11 time steps). The white inset denotes the morphology of the lung bud, while the dotted area denotes the segmented cell patch. Corresponding visual provided in *Figure 6—video 1*. Cells on both the apical and basal domains were 2.5D segmented, and their morphology quantified. Corresponding visual provided in *Figure 6—video 3*. Scale bars 20 µm. (**b**) Cell neighbour frequencies for the apical and basal layers over time. (**c**) Observed average neighbour number and area coefficient of variation (CV) for the apical and basal layers over time. (**d**) Growing apical and basal layers follow the AW law (black line). Colour

*Figure 6 continued on next page*

*Figure 6 continued*

code applies to e-f. (**e**) The relative average apical and basal cell areas are linearly related to the number of neighbours (in all time points) and follow Lewis' law (black line), or the quadratic relationship in the case of higher area variability (yellow line). (**f**) Observed fraction of hexagons versus area coefficient of variation (CV) on the apical and basal layers. The lines mark the theoretical prediction if polygonal cell layers follow either the linear Lewis' law (black line) or the quadratic law (yellow line).

The online version of this article includes the following video, source data, and figure supplement(s) for figure 6:

**Source data 1.** 2.5D neighbour number and area data over time.

**Source data 2.** 2.5D area CV and avg. neighbour number data over time.

**Source data 3.** 2.5D AW data over time.

**Figure supplement 1.** Lung bud viability and growth quantifications over time.

**Figure 6—video 1.** High-resolution light-sheet microscopy timelapse imaging of epithelial lung development (10 hr).

https://elifesciences.org/articles/68135#fig6video1

**Figure 6—video 2.** High-resolution light-sheet microscopy timelapse imaging of epithelial lung development (3 hr).

https://elifesciences.org/articles/68135#fig6video2

**Figure 6—video 3.** Apical and basal surface cell segmentations.

https://elifesciences.org/articles/68135#fig6video3

---

diameter of nuclei and the average apical cross-sectional area, there is insufficient space for all nuclei to be accommodated apically. Therefore, as a cell exits mitosis, the nucleus moves from the apical towards the basal side (G1 and S phase) and then back to the apical side (G2 phase) to undergo another round of mitosis, a process referred to as interkinetic nuclear migration (IKNM) (*Meyer et al., 2011*). Consequently, nuclei are distributed along the entire apical-basal axis, giving the tissue a pseudostratified configuration (*Norden, 2017*). We wondered to what extent the nuclear distribution, and its effect on the 3D cell shape, explains the observed area distributions and lateral T1 transitions.

To this end, we stained the nuclear envelope with fluorescently tagged antibodies against lamin B1 (*Figure 5a*, *Figure 5—figure supplement 1*), and 3D segmented all nuclei within epithelial cells in a tube segment (*Figure 5b*, *Figure 3—figure supplement 1*, *Figure 3—video 2*). The nuclei were distributed along the entire apical-basal axis (*Figure 5c*), and consistent with the pseudostratified appearance of the epithelium, nuclei in neighbouring cells had different positions along the apical-basal axis (*Figure 5b*). The nuclear shapes, volumes, and cross-sectional areas (*Figure 5d–f*) all varied along the apical-basal axis. As expected, nuclei are largest and most spherical at the apical side, where they undergo mitosis (*Figure 5d*). Thus, a one-sided, two-sample Welch t-test revealed a significantly reduced ellipticity of nuclei located in the first 25% of the apical-basal axis compared to those in the middle 50% (p=0.0002). The nuclear volumes are on average about 50% smaller than the cell volumes and largely correlate (r = 0.79, *Figure 5f*), likely reflecting parallel expansion during the cell cycle. Where present, the nuclear cross-sectional areas are only slightly smaller than those of the entire cell, and the cross-sectional areas of the cell and the nucleus are strongly correlated (r = 0.94, *Figure 5g*). The strong correlation can be accounted for by the opposing actions of cells and nuclei in the columnar epithelium. The nuclear volumes are too large to allow for a spherical nucleus to fit into a cylindrical cell of the measured height (*Figure 5h*). Accordingly, to fit into the cell, the nucleus necessarily has to deform. Nuclei respond to external forces with anisotropic shape changes (*Haase et al., 2016*; *Neelam et al., 2016*), which is consistent with the elliptical nuclear shapes that we observe (*Figure 5d*). However, there is a limit to how much the stiff nucleus can deform (*Lammerding, 2011*; *Shah et al., 2021*), resulting in a local widening of the cell where the nucleus is present. Cell sections without nucleus typically have smaller cross-sectional areas, thereby leading to a higher frequency of small cross-sections in cells compared to nuclei. Accordingly, as previously seen for the cell cross-sectional areas, the observed changes in cell neighbour numbers correlates with the observed changes in nuclear cross-sectional areas (*Figure 5i*) such that most T1L transitions occur where the nucleus starts and ends (*Figure 5j*).

We conclude that the positions of nuclei can explain much of the observed variability in the cross-sectional cell areas. During the cell cycle, nuclei migrate, and the cell volumes first increase, and

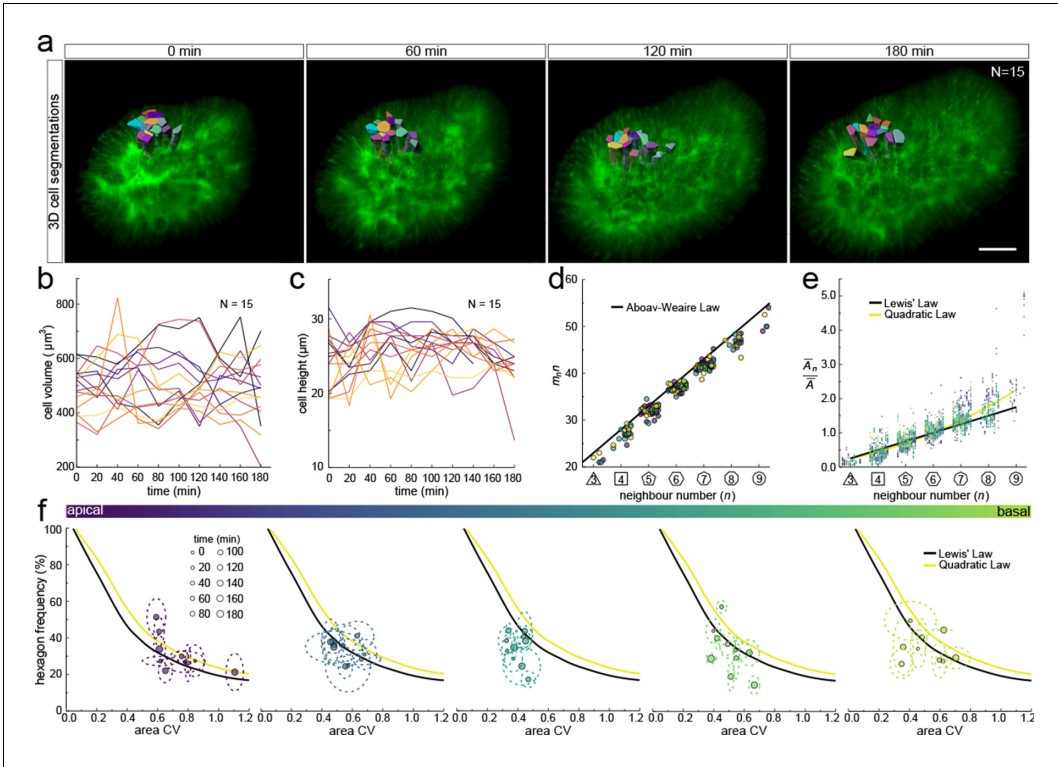

**Figure 7.** 3D cell organisation in growing epithelia. (**a**) 3D segmentation of 15 epithelial cells from timelapse light-sheet microscopy imaging of a mouse E12.5 distal lung bud expressing the Shh$^{GC/+}$; ROSA$^{mT/mG}$ reporter. The specimen was imaged every 20 min over 3 hr. Planar segmentations along the apical-basal axis were pooled into five groups to enable morphometric analysis in different tissue regions. A full timelapse panel is provided in *Figure 7—figure supplement 1* and *Figure 7—video 1*. Scale bar 30 µm. (**b**) Epithelial cell volume, and (**c**) height over time (N=15). (**d**) Segmented cells in pooled layers along the apical-basal axis follow the AW law (black line) over time (left to right); see panel f for colour code. (**e**) The relative average cell area in each layer is linearly related to the number of neighbours for all time points (left to right) and follows Lewis' law (black line), or the quadratic relationship in the case of higher area variability (yellow line); see panel f for colour code. (**f**) Temporal dynamics of observed fraction of hexagons versus area coefficient of variation (CV) along the apical-basal axis. Dotted lines denote variation per time point. Solid lines mark the theoretical prediction if polygonal cell layers follow either the linear Lewis' law (black line) or the quadratic law (yellow line).

The online version of this article includes the following video, source data, and figure supplement(s) for figure 7:

**Source data 1.** 3D neighbour number and area data over time.

**Source data 2.** 3D area CV and hex percent data over time.

**Source data 3.** 3D AW data over time.

**Figure supplement 1.** 3D timelapse segmentation of growing epithelia.

**Figure 7—video 1.** 3D timelapse segmentation of growing epithelia.

https://elifesciences.org/articles/68135#fig7video1

subsequently halve due to cell division. As all these processes affect the cross-sectional areas of the cells along the apical-basal axis, one would expect continuous spatial-temporal T1L transitions in growing pseudostratified epithelia.

## 3D cell organisation in growing epithelia

To follow 3D cellular dynamics during epithelial growth and deformation, we cultured embryonic lungs from a Shh$^{GC/+}$; ROSA$^{mT/mG}$ background and imaged every 20 min for a total of 10 hr using light-sheet microscopy (*Figure 6—video 1*). We used a subset of this dataset (11 time points, >3 hr) (*Figure 6—video 2*) to 2.5D segment the apical and basal surfaces, and to explore 3D cell shape dynamics and neighbour relationships in a growing lung bud (*Figure 6a*, *Figure 6—video 3*).

As the explant was growing, we readjusted the 2.5D segmented region such that the segmented surface area and cell numbers remained roughly constant over time (*Figure 6—figure supplement 1a*). Nonetheless, the segmented bud increased in volume as the thickness of the layer increased with time (*Figure 6—figure supplement 1b*). Much as in the static dataset, the neighbour number distributions (*Figure 6b*), and variability of cross-sectional areas (*Figure 6c*) differ between the apical and basal cell layers in all time points. However, for all time points, both the apical and basal layers conformed to Euler's polyhedron formula (*Figure 6c*), Aboav-Weaire's law (*Figure 6d*), and Lewis' law (*Figure 6e*). Furthermore, the fraction of hexagons also followed from the variability of the cross-sectional areas, as predicted by the theory (*Figure 6f*).

We next sought to analyse the 3D dynamics of segmented epithelial cells. As the tracking of packed cells in growing pseudostratified epithelia is challenging, we focused on a small patch with 15 cells in total (*Figure 7a*). Sequential contour surfaces were drawn to follow cell membrane outlines on several planes along the apical-basal axis and interpolated to reconstruct 3D morphology for each time point (*Figure 7—figure supplement 1*, *Figure 7—video 1*). All planar segmentations were then pooled into five groups along the apical-basal axis to enable morphometric analysis in different tissue regions. Over the time course, the volume of individual cells varied between roughly 400 and 800 µm³ (*Figure 7b*), and the apical-basal length varied between roughly 20 and 30 µm (*Figure 7c*). We note that all layers conform to Aboav-Weaire's law (*Figure 7d*) and Lewis' law (*Figure 7e*) in all time points. Moreover, consistent with our theory, the fraction of hexagons follows from the variability of the cross-sectional area, though more deviations are observed, given the small number of cells analysed (*Figure 7f*).

Much as in the static dataset (*Figures 3* and *4*), we observe up to 14 neighbour number changes (T1L transitions) along the apical-basal axis (*Figure 8a,b*). The average number of T1L transitions is relatively constant over time (*Figure 8b*). The mean relative apical-basal position for T1L transitions is again roughly in the middle, but in this small dataset, we now observe more T1L transitions in the center of the cell than at the apical or basal boundaries (*Figure 8c*). By following a single cell over time, we can appreciate the dynamic cell shape changes, and how a change in the cross-sectional area correlates with a change in neighbour number (*Figure 8d*). The neighbour relationships are, of course, not determined by the local cell cross-section, but by the overall cross-sectional area distribution in that layer. Accordingly, the correlation between the cross-sectional area and the neighbour number is not perfect for a single cell. By considering a patch of cells, we can, however, see how those T1L transitions occur dynamically in developing tissues (*Figure 8e*).

## Discussion

Epithelial tissues remodel into complex geometries during morphogenesis. We used light-sheet microscopy and 3D cell segmentation to unravel the physical principles that define the 3D cell neighbour relationships in pseudostratified epithelial tissues. Our analysis reveals that pseudostratified epithelial layers adopt a far more complex packing solution than previously anticipated: the 3D epithelial cell shapes are highly irregular, and cell neighbour relationships change multiple times along the apical-basal axis, with some cells having up to 14 changes in their neighbour contacts along their apical-basal axis (*Figure 3a*). Curvature effects can result in neighbour changes, but the data does not show the dependency on cell neighbour numbers that would be expected if curvature effects played a dominating role (*Figure 3g*). There is also no apical-basal bias (*Figure 3e*), and the prevalence of contact remodelling is randomly distributed (*Figure 4i*).

Even though the neighbour relationships are uncorrelated between the apical and basal sides and appear random at first sight, they follow the same fundamental relationships that have previously been described for apical epithelial layers, that is, Euler's polyhedron formula, Lewis' law, and Aboav-Weaire's law across the entire tissue and at all times (*Figures 2*, *4*, *6* and *7*). This arrangement minimises the lateral cell-cell surface energy in each plane along the apical-basal axis, given the variability in the cell cross-sectional areas (*Figure 1*; *Kokic et al., 2019*; *Vetter et al., 2019*). Where present, the stiff nucleus determines the cell cross-sectional area, as is apparent from the strong correlation between the cell cross-sectional and the nuclear cross-sectional areas (*Figure 5g*). Accordingly, most changes in neighbour relationships occur at the apical and basal limits of the nucleus where cross-sectional areas change sharply (*Figure 5i*). As the nucleus moves along the apical-basal axis during the cell cycle, a process referred to as interkinetic nuclear migration (IKNM)

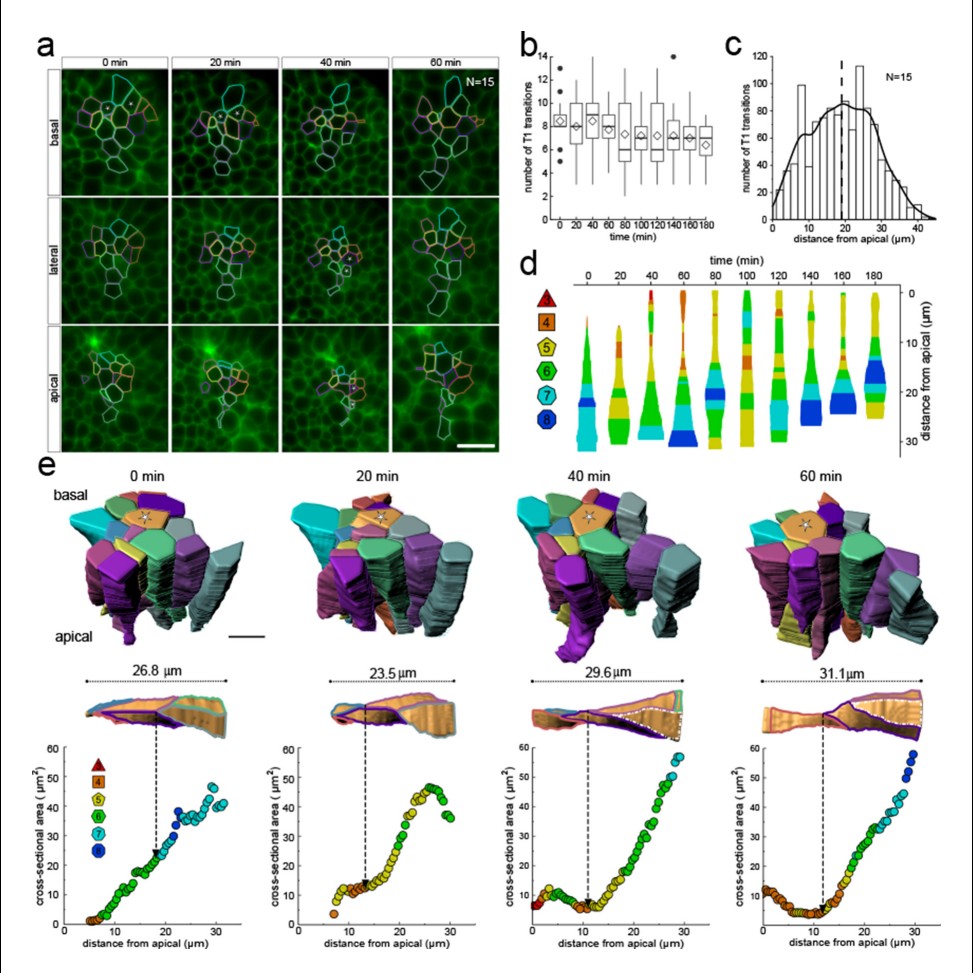

**Figure 8.** 3D cell neighbour dynamics in growing epithelia. (**a**) Apical, basal and lateral cross-sections from a light-sheet microscopy timelapse of a murine E12.5 distal lung bud expressing the Shh[GC/+]; ROSA[mT/mG] reporter. The specimen was imaged every 20 min over 3 hr. Cell membrane outlines illustrate fluid cell neighbour relationships along the apical-basal axis and over time. T1 transitions are marked with white stars; see panel e for cell colour code. Scale bar 14 µm. (**b**) Number of T1L transitions for all cells (N=15) over time. Diamonds represent the mean. Morphometric quantifications of planar segmentations along the apical-basal axis were used to examine T1L transition dynamics. (**c**) Spatial distribution along the apical-basal axis of T1L transitions for all cells and time points. (**d**) Temporal evolution of neighbour relationships along the apical-basal axis for a single cell. Schematic cell width corresponds to cross-sectional area. (**e**) (top row) 3D iso-surface segmentations of 15 epithelial cells. Scale bar 10 µm. (bottom row) Cross-sectional area and cell neighbour number along the apical-basal axis for a given cell (marked with a white star). Dotted lines indicate contact with a cell that was not segmented.

(*Meyer et al., 2011*), cell neighbour relationships change continuously (*Figure 8*). We conclude that neighbour relationships in epithelia are fluidic, and the complex, dynamic 3D organisation of cells in growing epithelia follows simple physical principles.

Defining the physical principles behind cell neighbour relationships is only the first step in unravelling the determinants of epithelial 3D cell shapes. The second key aspect is the cell volume distribution along the apical-basal axis, which gives rise to the cell cross-sectional area distribution, which then determines the cell neighbour relationships (*Figure 5e*). The overall cell volume is determined by cell growth and division, but its distribution along the apical-basal axis depends on the nuclear dynamics (*Figure 5g*), and the epithelial cell heights. We find that the nucleus occupies, on average, 55% of the cell volume in the embryonic lung epithelia. As the cell nuclei move along the apical-basal axis during the cell cycle (*Meyer et al., 2011*), the cytoplasm fills the remaining space between the apical and basal surfaces, likely in a way that minimises the total surface area of all cells. The

determinants of the epithelial thickness, that is, the distance between the apical and basal surfaces are still unknown, but signalling downstream of Fibroblastic Growth Factor (FGF), Sonic Hedgehog (SHH), Bone Morphogenetic Protein (BMP)/ transforming growth factor-beta (TGF-β), and WNT has been observed to affect cell height, presumably via an effect on cell tension and/or cell-cell adhesion (*Gritli-Linde et al., 2002*; *Hirashima and Matsuda, 2021*; *Kadzik et al., 2014*; *Kondo and Hayashi, 2015*; *Widmann and Dahmann, 2009*).

Cell-based modelling frameworks are heavily used to investigate epithelial processes and how they result in morphological changes such as tissue bending, folding, fusion, and anisotropic growth during morphogenesis (*Fletcher et al., 2014*; *Tanaka, 2015*). Our data confirms many underlying assumptions of cell-based modelling frameworks and provides quantitative data to calibrate parameters. Once calibrated to reproduce the here identified 3D cell shape distributions, such simulation frameworks will help to reveal the determinants of 3D cell shapes, and will be invaluable in providing insight into how local changes in cell growth, adhesion, tension, or in the basal lamina affect cell shapes locally and within the remaining epithelial layer.

In summary, this study offers a detailed view of 3D cell neighbour relationship dynamics and packing in growing epithelial tissues, and demonstrates that the 3D cell shapes are much more complex than previously anticipated, and that cell neighbour relationships are dynamic and change as result of cell growth and cell cycle-linked IKNM. The complex 3D cell neighbour relationships can nonetheless be understood based on simple physical principles. Although we recognize that tissue architecture is a multifactorial process, our work carries vast implications for the study of cell-cell signalling, epithelial cohesion, and energetic modelling of developing epithelial layers in both healthy and disease contexts.

## Materials and methods

### Ethical statement

Permission to use animals was obtained from the veterinary office of the Canton Basel-Stadt (license number 2777/26711). Experimental procedures were performed in accordance with the Guide for the Care and Use of Laboratory Animals and approved by the Ethics Committee for Animal Care of ETH Zurich. All animals were housed at the D-BSSE/UniBasel facility under standard water, chow, enrichment, and 12 hr light/dark cycles.

### Animals

To investigate 3D cellular dynamics during mouse embryonic lung development, we used mouse lung rudiments from animals homozygous for the $Rosa26^{mTmG}$ and heterozygous for the $Shh^{GFP-Cre}$ allele ($Shh^{GFP-Cre/+}$; $Rosa26^{mTmG}$). The double-fluorescent *Shh*-controlled *Cre* reporter mouse expresses membrane-targeted tandem dimer Tomato (mT) before CRE-mediated excision and membrane-targeted green fluorescent protein (mG) after excision (*Muzumdar et al., 2007*). As a result, only epithelial cell membranes are labelled by GFP, while all adjacent mesenchymal tissue is labelled by tdTomato.

### Immunofluorescence

E12.5 mouse lungs were fixated for 1 hr in 4% paraformaldehyde in PBS, and subsequently incubated with Lamin B1 (Thermo; Material No. 702972; 1:200) at 4°C for 3 days. As a structural component of the nuclear lamina, LaminB1 immunostaining makes crowded nuclei clearly distinguishable and easily segmentable. After washing in D-PBS, lungs were incubated with conjugated fluorescent secondary Alexa Fluor 555 donkey anti-mouse IgG (H+L) (Abcam; Material No. ab150106; 1:250) for 2 days at 4°C.

### Optical clearing and lightsheet imaging

Optical clearing of embryonic lung rudiments enabled the 3D segmentation of numerous epithelial cells from single image stacks. To this extent, the whole-mount clearing of dissected E12.5 lung explants was performed with the Clear Unobstructed Brain/Body Imaging Cocktails and Computational Analysis (CUBIC) protocol (*Susaki et al., 2015*; *Figure 2—figure supplement 3*). Reagents for delipidation and refractive index (RI) matching were made as follows: CUBIC-1 [25% (w/w) urea, 25%

ethylenediamine, 15% (w/w) Triton X-100 in distilled water], and CUBIC-2 [25% (w/w) urea, 50% (w/w) sucrose, 10% (w/w) nitrilotriethanol in distilled water], respectively. Following fixation and immunostaining, samples were incubated in 1/2 CUBIC-1 (CUBIC-1:H2O=1:1) for four days, and in 1X CUBIC-1 until they became transparent. All explants were subsequently washed several times in PBS and treated with 1/2 CUBIC-2 (CUBIC-2:PBS=1:1) for around four days. Lastly, incubation in 1X CUBIC-2 was done until the desired transparency was achieved. All solutions were changed daily, and CUBIC-1 steps were performed on a shaker at 37°C while CUBIC-2 steps at room temperature. Cleared samples were then embedded in 2% low melting point solid agarose cylinders and immersed in CUBIC-2 for two more days to increase the agarose refractive index. 3D image stacks were acquired on a Zeiss Lightsheet Z.1 microscope using a Zeiss 20x/1.0 clearing objective .

## Timelapse light-sheet acquisitions

Light-sheet acquisitions of live epithelial cell morphology enabled the study of 3D organisation dynamics. Following dissection in DPBS at room temperature, E12.5 lung explants were cultured in sterile Dulbecco's modified Eagle's medium w/o phenol red (DMEM) (Life Technologies Europe BV; 11039021) containing 10% Fetal Bovine Serum (FBS) (Sigma-Aldrich Chemie GmbH; F9665-500ML), 1% Glutamax (Life Technologies Europe BV; A1286001), and 1% penicillin/streptomycin (Life Technologies Europe BV; 10378–016). All specimens were equilibrated at 37°C with 5% CO2 in a humidified incubator for 1 hr.

Following a 1 hr equilibration period, 1.5% LMP hollow agarose cylinders were prepared (*Udan et al., 2014*). Hollow cylinders, in contrast to solid ones, accommodate unencumbered 3D embryonic growth, provide boundaries to minimise tissue drift, enable imaging from multiple orientations, and allow for better perfusion of gasses and nutrients. All specimens were suspended within each hollow cylinder in undiluted Matrigel (VWR International GmbH; 734–1101), an ECM-based optically clear hydrogel that provided a near-native 3D environment and supported cell growth and survival. All cylinders were kept at 37°C with 5% $CO_2$ in culture media for 1 hr before mounting.

For each overnight culture, the imaging chamber was prepared by sonication at 80°C, followed by ethanol and sterile PBS washes. After assembly, the chamber was filled with culture medium and allowed to equilibrate at 37°C with 5% $CO_2$ for at least 2 hr before a cylinder containing an explant was mounted for imaging. Furthermore, to compensate for evaporation and to maintain a fresh culture media environment, two peristaltic pumps were installed to supply 0.4 mL and extract 0.2 mL of culture medium per hour. Each lung explant was then aligned with the focal plane within the center of a thin light-sheet to enable fine optical sectioning with optimal lateral resolution. For this study, all live imaging was done with a 20x/1.0 Plan-APO water immersion objective.

## Image processing

To efficiently process the resulting volumetric CZI datasets (10s-100s of GBs), all image stacks were transferred to a storage server and subsequently processed in remote workstations (Intel Xeon CPU E5-2650 with 512 GB memory). Deconvolution via Huygens Professional v19.04 (Scientific Volume Imaging, The Netherlands, http://svi.nl) improved overall contrast and resolution while Fiji (ImageJ v1.52t) (*Schindelin et al., 2012*) aided in accentuating cell membranes, enhancing local contrast, removing background fluorescence, and TIFF conversion.

## Cell morphometric quantifications

Cell morphology on the apical and basal membranes of embryonic lung epithelia was investigated using the open-source software platform MorphoGraphX (MGX) (*Barbier de Reuille et al., 2015*). By meshing the curved boundaries of input 3D image stacks and projecting nearby signal onto it, MGX builds a curved 2.5D image projection that is distortion-free, unlike planar 2D projections that ignore curvature. We then proceeded to use a suitable implementation of the Watershed transform to extract individual cell geometries, with minimal manual curation, and quantify properties such as surface area and the number of cell neighbours. All border cells were excluded. Apical and basal cell meshes were exported as text files and traversed using the R Programming Environment to extract the neighbour relationships between cells as needed to generate Aboav-Weaire plots (*Gómez et al., 2021*).

To render time-lapse datasets and extract 3D volumetric surface reconstructions of entire epithelial cells, we employed Imaris v9.1.2 (Bitplane, South Windsor, CT, USA). By computationally interpolating between cell membrane contour surfaces from successive transverse frames into iso-surfaces, faithful cell and nuclear 3D volumes were obtained. Quantified volumetric features included cell and nuclear volume, total surface area, sphericity, and nuclear position along the apical-basal axis. Imaris was also used to generate high-resolution videos, which, despite being strongly downsampled to accommodate vast time-lapse datasets, presented little noticeable loss in image quality. Furthermore, to extract cell areas and the number of neighbours along the apical-basal axis, transverse image frames were imported into ImageJ and processed using the interactive plugin TissueAnalyzer (*Aigouy et al., 2016*). Like this, cell segmentation masks across layers could be generated, and cell geometry and neighbour topology quantified (*Gómez et al., 2021*).

## Acknowledgements

This work has been supported through an SNF Sinergia grant to DI. We acknowledge the many discussions and varied input of Marco Kokic, Anđela Markovic, Odyssé Michos, and Erand Smakaj, and we thank Richard Smith for help with MorphographX.

## Additional information

### Funding

| Funder | Grant reference number | Author |
| --- | --- | --- |
| Schweizerischer Nationalfonds zur Förderung der Wissenschaftlichen Forschung | CRSII5_170930 | Dagmar Iber |

The funders had no role in study design, data collection and interpretation, or the decision to submit the work for publication.

### Author contributions

Harold Fernando Gómez, Data curation, Software, Formal analysis, Supervision, Validation, Investigation, Visualization, Methodology, Writing - original draft, Writing - review and editing; Mathilde Sabine Dumond, Data curation, Software, Formal analysis, Funding acquisition, Validation, Investigation, Visualization, Methodology; Leonie Hodel, Formal analysis, Investigation; Roman Vetter, Software, Formal analysis, Supervision, Investigation, Visualization, Methodology, Writing - original draft, Writing - review and editing; Dagmar Iber, Conceptualization, Formal analysis, Supervision, Funding acquisition, Investigation, Methodology, Writing - original draft, Project administration, Writing - review and editing

### Author ORCIDs

Harold Fernando Gómez ⬤ https://orcid.org/0000-0001-5200-4933
Leonie Hodel ⬤ https://orcid.org/0000-0002-2541-4133
Roman Vetter ⬤ https://orcid.org/0000-0003-2901-7036
Dagmar Iber ⬤ https://orcid.org/0000-0001-8051-1035

### Ethics

Animal experimentation: Permission to use animals was obtained from the veterinary office of the Canton Basel-Stadt (license number 2777/26711). Experimental procedures were performed in accordance with the Guide for the Care and Use of Laboratory Animals and approved by the Ethics Committee for Animal Care of ETH Zurich.

### Decision letter and Author response

Decision letter https://doi.org/10.7554/eLife.68135.sa1
Author response https://doi.org/10.7554/eLife.68135.sa2

## Additional files

**Supplementary files**

• Transparent reporting form

### Data availability

The source code and plotted data files are available as a git repository at https://git.bsse.ethz.ch/iber/Publications/2021_gomez_3d_cell_neighbour_dynamics.git. The raw data is publicly available as openBIS repository at https://openbis-data-repo.ethz.ch/openbis/webapp/eln-lims/?user=observer&pass=openbis under the name 3D Epithelium.

The following dataset was generated:

| Author(s) | Year | Dataset title | Dataset URL | Database and Identifier |
|---|---|---|---|---|
| Gómez HF, Dumond MS, Hodel L, Vetter R, Iber D | 2021 | 3D Epithelium | https://openbis-data-repo.ethz.ch/openbis/webapp/eln-lims/?user=observer&pass=openbis | openBIS, 3D Epithelium |

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

## Appendix 1

## A simple theory for lateral T1 transitions in curved tissues

Drawing inspiration from the notion of 'scutoids' (*Gómez-Gálvez et al., 2018*), we develop here a simple geometrical theory to estimate the effect of tissue curvature, if any, on the occurrence and number of lateral T1 transitions of cells along the apical-basal axis in a pseudostratified epithelium. The purpose of this effort is to verify whether an idea brought forward in the aforementioned article, which is that tissue curvature could be responsible for the occurrence of up to one lateral T1 transition per cell between its apical and basal sides, is consistent with our observations in the developing, pseudostratified mouse lung epithelium.

We start by considering, like the authors of *Gómez-Gálvez et al., 2018*, the vicinity of an edge shared by two adjacent cells in a two-dimensional cross section of the tissue perpendicular to the apical-basal axis. This vicinity is defined as the quadrilateral (*Appendix 1—figure 1A*, orange) spanned by the four cell junctions (green) that are directly linked to the edge of interest (blue), covering a fraction of four cells in such a two-dimensional projection. As the projection plane is moved along the apical-basal axis, the edge length can shrink to zero and reappear with different orientation, leaving the cell neighbourhood of all four cells modified by one. It is this process which we refer to as lateral neighbourhood transition of type T1. Each such edge vicinity is characterised by an aspect ratio $\epsilon = w/h$, where $w = (w_1 + w_2)/2$ is the average width and $h = (h_1 + h_2)/2$ the average height. We now proceed to quantitatively estimate with a simple geometrical model how a change in tissue curvature translates into a change of aspect ratio of such motifs, and how that in turn would be expected to change the cell neighbourhood if all T1 transitions were effectively governed by the geometric effect of changing tissue curvature.

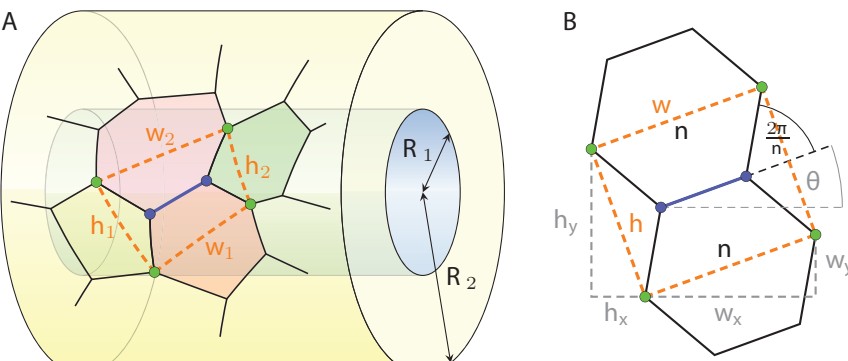

**Appendix 1—figure 1.** Definition of the vicinity of a cell edge in a geometric projection on a tubular epithelium.

We consider two different tissue topologies: (i) a cylindrical tube and (ii) a spherical vesicle or hemispherical tube cap. The crucial difference between these two cases is that the former only has a mean curvature but no Gaussian curvature, whereas the latter has both. As the radius $r$ of these surfaces changes, the change of aspect ratio follows by the quotient rule of calculus as

$$\frac{\partial \epsilon}{\partial r} = \frac{\partial (w/h)}{\partial r} = \frac{1}{h}\left(\frac{\partial w}{\partial r} - \epsilon \frac{\partial h}{\partial r}\right). \tag{5}$$

We now consider the general case where the edge motif is arbitrarily oriented on the surface, and will later average over all possible orientations. In general, the edge of interest spans an angle $\theta$ with the first principal curvilinear axis of the surface, which we take as the main cylinder axis for the tubular epithelium (*Appendix 1—figure 1*), and as any great circle on the spherical epithelium. The change of width and height then read

$$\begin{aligned}
\frac{\partial w}{\partial r} &= \cos\theta \frac{\partial w_x}{\partial r} + \sin\theta \frac{\partial w_y}{\partial r}, \\
\frac{\partial h}{\partial r} &= \sin\theta \frac{\partial h_x}{\partial r} + \cos\theta \frac{\partial h_y}{\partial r},
\end{aligned} \tag{6}$$

where $w_x$, $w_y$ and $h_x$, $h_y$ are the components of the width and height vectors, respectively (**Appendix 1—figure 1B**, gray):

$$
\begin{aligned}
w_x &= w\cos\theta, & w_y &= w\sin\theta, \\
h_x &= h\sin\theta, & h_y &= h\cos\theta.
\end{aligned}
\tag{7}
$$

Their response to a change in tissue radius is given by

$$
\begin{aligned}
\frac{\partial w_x}{\partial r} &= \kappa_1 w_x, & \frac{\partial w_y}{\partial r} &= \kappa_2 w_y, \\
\frac{\partial h_x}{\partial r} &= \kappa_1 h_x, & \frac{\partial h_y}{\partial r} &= \kappa_2 h_y,
\end{aligned}
\tag{8}
$$

where $\kappa_1$ and $\kappa_2$ are the principal curvatures of the surface in the respective directions. Substituting **Equation 8** into **Equations 7, 6 and 5** yields

$$
\frac{\partial \epsilon}{\partial r} = \epsilon(\kappa_1 - \kappa_2)\cos(2\theta).
$$

A cylinder with radius $r$ has the principal curvatures $\kappa_1 = 0$ and $\kappa_2 = 1/r$, so

$$
\frac{\partial \epsilon}{\partial r} = -\frac{\epsilon}{r}\cos(2\theta).
$$

For a sphere with radius $r$, on the other hand, the two principal curvatures are both equal, $\kappa_1 = \kappa_2 = 1/r$, and thus the aspect ratio does not change with varying radius,

$$
\frac{\partial \epsilon}{\partial r} = 0
$$

independent of the orientation $\theta$.

From here onward, we consider the region along the apical-basal axis between two T1 transitions on a given cell, irrespective of whether these T1 transitions increase or decrease the neighbour number. In our experimental quantifications of lateral T1 transition in the mouse lung epithelium, they were counted in the same way. The sign of the T1 transition is encoded in the sign of $\partial\epsilon/\partial r$, hence we consider only the absolute value $|\partial\epsilon/\partial r|$ hereafter, to quantify the change of edge aspect ratio between T1 transitions along the tissue radius.

Carrying on with the tubular geometry, we now average over all orientations $\theta$ to calculate the mean change of aspect ratio for the entire cell ensemble in the tissue. The set of possible orientations is $\theta \in [0, \pi/2)$, because all angles $\theta \in [\pi/2, \pi)$ just switch the roles of $w$ and $h$. Therefore,

$$
\left|\frac{\partial \epsilon}{\partial r}\right| = \frac{2}{\pi}\int_0^{\pi/2}\left|\frac{\epsilon}{r}\cos(2\theta)\right|d\theta = \frac{2\epsilon}{\pi r}.
$$

This is an ordinary differential equation which can easily be solved by separation of variables. The result is

$$
\epsilon(r) = \epsilon_1\left(\frac{R_1}{r}\right)^{\pm 2/\pi}
$$

and therefore

$$
\left|\frac{\partial \epsilon}{\partial r}\right| = \frac{2\epsilon_1}{\pi r}\left(\frac{R_1}{r}\right)^{\pm 2/\pi}
\tag{9}
$$

where $\epsilon_1$ is the aspect ratio at a given cylinder radius $R_1$. For the sphere, on the other hand, we trivially have a constant aspect ratio $\epsilon(r) = \epsilon_1$.

To link this change of aspect ratio with lateral T1 transitions, a relationship between $\epsilon$ and the number of neighbours a cell locally has, $n$, needs to be established. A simple rough estimate can be found by recalling that cells typically tend toward a regular polygonal shape in a cross-sectional projection (**Kokic et al., 2019**; **Vetter et al., 2019**). In a honeycomb lattice, the aspect ratio is

$$\epsilon|_{n=6} = \frac{2}{\sqrt{3}} \approx 1.155.$$

For general regular polygons with $n \geq 4$, the relationship reads

$$\epsilon(n) = \frac{1 + 2\cos(2\pi/n)}{2\sin(2\pi/n)} \tag{10}$$

as can be recognised from **Appendix 1—figure 1B**. **Equation 10** holds locally in the region between two consecutive T1 transitions, where $n$ is constant. From this, we can estimate the change of aspect ratio between two lateral T1 transition as

$$\frac{\partial\epsilon}{\partial n} = \pi \frac{2 + \cos(2\pi/n)}{n^2 \sin^2(2\pi/n)}. \tag{11}$$

On average, for hexagonal cells ($n = 6$), this evaluates to

$$\frac{\partial\epsilon}{\partial n}\bigg|_{n=6} = \frac{5\pi}{54} \approx 0.291.$$

So how many T1 transitions can be expected along the apical-basal axis? We can apply the chain rule of calculus to write

$$\left|\frac{dn}{dr}\right| = \left|\frac{\partial n}{\partial\epsilon}\frac{\partial\epsilon}{\partial r}\right| = \left|\left(\frac{\partial\epsilon}{\partial n}\right)^{-1}\frac{\partial\epsilon}{\partial r}\right|. \tag{12}$$

Substituting **Equations 9 and 11** into **Equation 12**, we find

$$\left|\frac{dn}{dr}\right| = \frac{n^2 \sin^2(2\pi/n)}{2 + \cos(2\pi/n)}\frac{2\epsilon_1}{\pi^2 r}\left(\frac{R_1}{r}\right)^{\pm 2/\pi}$$

for a cylindrical epithelium, and $dn/dr = 0$ for a spherical one. A spherical topology is thus entirely compatible with cells being frusta that do not change their neighbour number from their apical to the basal side, whereas a cylindrical topology is not. For tubes, a change of radius $R_1 \rightarrow R_2$ lets the neighbour number change by

$$|\Delta n| = \int_{R_1}^{R_2}\left|\frac{dn}{dr}\right|dr = \frac{\epsilon_1}{\pi}\frac{n^2 \sin^2(2\pi/n)}{2 + \cos(2\pi/n)}\left[1 - \left(\frac{R_1}{R_2}\right)^{\pm 2/\pi}\right]$$

per cell edge. How far must $R_1$ and $R_2$ be apart to expect one T1 transition per cell? Over that distance, each of the cell's $n$ edges will have undergone a change of $1/n$ on average. The answer can therefore be found by setting $|\Delta n| = 1/n$ with $\epsilon_1 = \epsilon(n)$ from **Equation 10**, and solving for the radius fold change:

$$\frac{R_2}{R_1} = \left(1 - \frac{\epsilon'(n)}{n\epsilon(n)}\right)^{\pm\pi/2} = \left(1 - \frac{2\pi[2 + \cos(2\pi/n)]}{n^2[\sin(2\pi/n) + \sin(4\pi/n)]}\right)^{\pm\pi/2} \tag{13}$$

This is the expected fold change of mean curvature inside a tubular epithelium between two consecutive lateral T1 transitions per cell, if the T1 transitions were mainly driven by a change of curvature. Note that the two cases with different signs in the exponent reflect the $R_1 \leftrightarrow R_2$ symmetry. As shown in **Appendix 1—figure 2**, **Equation 13** predicts a strong dependency of the required change of curvature between adjacent lateral T1 transitions on the number of neighbours the cell has, $n$. Cells with a large neighbour number are expected to transition at much smaller changes of curvature on average than cells with few neighbours.

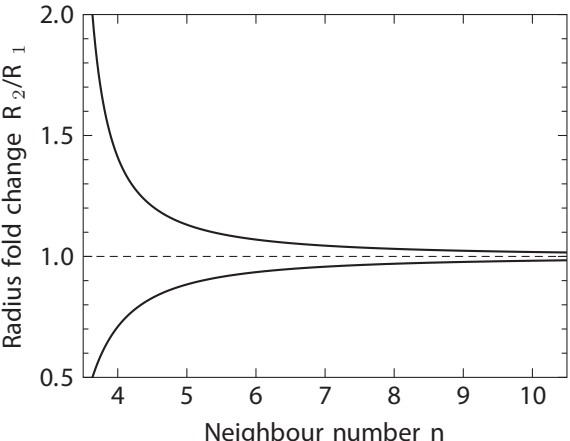

**Appendix 1—figure 2.** Predicted radius fold change in a tubular epithelium between a pair or consecutive T1 transitions per cell.

*Equation 13* relates the fold change of tissue curvature on a tubular sheet *between* consecutive lateral T1 transitions along the apical-basal axis of a single cell, to the cell's neighbour number $n$ in any cross section perpendicular to the radius between the two transitions. $n$ here is to be understood *locally within the tissue*, not as the cell connectivity on either the apical or basal surfaces of the cell. *Equation 13* does not directly lend itself to estimating the number of lateral T1 transitions to be expected for cells in a curved tissue, although our theory might be adaptable to this case with some modifications. The sign of T1 transitions is ignored here, leaving the possibility of frequent back-and-forth transitions with zero net effect on $n$—a phenomenon that we indeed do observe in the developing lung epithelium. *Equation 13* quantifies only the net contribution of curvature on the number of T1 transitions a cell undergoes. This net effect strongly depends on the the local neighbour number $n$, as shown in *Appendix 1—figure 2*.

