## [Decision Letter]

**Acceptance summary:**

In this study, the authors measured the three-dimensional organization within an epithelial cell monolayer and found that cell neighbors change frequently along the apicobasal axis. This work constitutes a significant investigation into the drivers of complex three-dimensional cell shapes and tissue structures and will be of interest to scientists studying tissue mechanics.

**Decision letter after peer review:**

Thank you for submitting your article "3D Cell Neighbour Dynamics in Growing Pseudostratified Epithelia" for consideration by *eLife*. Your article has been reviewed by 3 peer reviewers, one of whom is a member of our Board of Reviewing Editors, and the evaluation has been overseen by Anna Akhmanova as the Senior Editor. The following individual involved in review of your submission has agreed to reveal their identity: Matthias Merkel (Reviewer #3).

Essential revisions:

1) Claims that imply causation but are based on correlations need to be either substantiated further to justify causation or toned down. Reporting only correlations might not suffice for publication in *eLife*.

2) Solve the issues spotted by Reviewer #3 in the theory presented in the SM (reviewer's point 4).

3) Estimate uncertainties as requested by Reviewer #3, point 3.

4) Respond to the recommendations for the authors.

*Reviewer #1 (Recommendations for the authors):*

L 60: Eq. (2) is wrong and should read instead 5+8/n.

Figure 1: I did not understand, why the authors reproduce the previously published data in Figure 1e-k. Figure 1d does not seem to be referred to in the text.

Figure 2f: It is hard to distinguish between linear and quadratic dependences.

Figure 3e: For tips there seems a clear bias for apical transitions. Could you comment on this?

Ll 165: I did not understand this part. Why do you fit ellipses and not circles? Why do you consider two subsequent T1L transitions/consecutive neighbor number changes and not one?

L 211: This statement contradicts later statements that involve the nucleus. Also, here you really write about minimizing a line energy, whereas it should be a surface energy; the lateral and the apico-basal directions are not independent.

L 233: What do you mean by "pseudo-stratification"? Where is it shown?

Ll 241+ Figure 5d: There seems to be only a very weak correlation between nuclear volume and ellipticity or position along the apico-basal axis.

Ll 244 + Figure 5f,g: Since the nuclei are inside the cell, could it be otherwise? Also, the correlation between cell and nuclear volume seems to be rather elevated. Is this correct?

Ll 246: But it could also be the other way around or some independent factor could determine both. I do not think that the consistency you notice has any explanatory power.

Figure 6: Are the results based on one explant?

L 317: "Where present, the stiff nucleus determines the cell cross-sectional area, as is apparent from the strong correlation between the cell cross-sectional and the nuclear cross-sectional areas" Correlations do not imply causation. I do not think that you have data to support the causal relationship.

L 320: Again, this seems to be speculation. Figure 8 does not show the nucleus – and even if: from your the data it is not clear that the motion of the nucleus causes changes in the neighbor relationships.

*Reviewer #2 (Recommendations for the authors):*

Page 4, line 73, "following, Aboav-Weaire's law": I suggest replacing this with "following Aboav-Weaire's law" (i.e. remove the comma).

Page 4, lines 89-90, "As such, growth and cell division determine the variability of the apical areas" and lines 100-101, "The cross-sectional areas vary as a result of cell growth, division and interkinetic nuclear migration (IKNM)": Maybe I missed it, but can the authors discount any role(s) for 'active' cell intercalations and/or cell death here?

Page 4, lines 97-98, "the variation of […] define the": I suggest replacing this with "the variation of […] axis defines the" (i.e. add an 's').

Page 5, lines 112-113, "We detected less than half as many cells on the apical side": Just to be clear here, do the authors mean that they could only detect half as many cells on the apical side due to their small size and/or lower resolution, or that there are really half as many cells on the apical side and hence over half the cells don't extend fully from the basal side to the apical side?

Page 5, lines 118-119, "the neighbour changes in spherical geometries cannot be explained with such an effect (Gómez-Gálvez et al., 2018)": For clarity, I suggest briefly emphasizing why not (I guess the point is that as one looks along the apical-basal axis in this case, one is just seeing a dilation, so prismatic cells do the job fine).

Page 5, lines 127-130, "Aboav-Weaire's law (Figure 2e) and Lewis' law (Figure 2f) hold not only for the apical, but also for the basal datasets […] the apical layers, which have a larger area variability than the basal layers (Figure 2c), follow the quadratic law (yellow line) rather than the linear Lewis' law (black line)": I may simply have misunderstood things here, but the first sentence seems to indicate that Lewis' law holds for the apical dataset, while the second sentence seems to indicate that the apical layers follow the quadratic law rather than Lewis' law, which seems a contradiction. If I have indeed just misunderstood things here, then perhaps the authors could slightly rephrase these sentences for clarity.

Page 7, lines 232-233, "giving the tissue a pseudostratified configuration": I suppose strictly speaking this is true of pseudostratified epithelia but not ALL epithelia, so it might be worth changing "In epithelia" to "In many epithelia" on line 227, or similar.

Page 9 lines 335-336, "signalling factors that control cellular tension are known to affect cell height (Widmann and Dahmann, 2009)": For clarity, perhaps the authors could name these signalling factor(s) explicitly here?

Page 16, Figure 1, panel (b): The chosen colour scheme makes it very difficult for me to tell the difference between 5, 6, 7 and 8 sided cells, since they're all shades of green. If possible, it would be great if the authors could use more contrasting colours.

Page 16, Figure 1, panel (e): Can the authors briefly explain in the legend what the various abbreviations mean (cNT, EYE, dPE, dWL, dWP, dMWP, dPW, TP, dMWL), or at least broadly which different tissues/species are considered?

Page 19, Figure 3, panel (b): I found this figure difficult to understand partly due to the colour scheme but also because the authors are plotting two datasets on the same image. If there is a way of simplifying this figure at all, that would be great.

*Reviewer #3 (Recommendations for the authors):*

– Citation of Rupprecht et al., MBoC, 2017 appears justified, as it is the first / one of the first papers to discuss apico-basal T1 transitions in epithelia based on experimental data.

– Recommendations regarding point 3. in the public review:

a) Estimations of uncertainties (e.g. standard error of the mean) would be useful, e.g. in Figure 2 e,f. For instance, in Figure 2f, an indication for the stochastic uncertainty due to the finite sample size would be useful.

b) Aboav-Weaire law: For instance in Figure 2e, everything is close to a diagonal line. However, that there is a line with slope of approximately 6 in the plots of m_n*n over n is already a consequence of the Euler characteristic. To better display the correlation that is the core of the Aboav-Weaire law, I would suggest to directly plot m_n over n (as also one e.g. in Aboav, 1970). Together with error bars indicating uncertainty, this would allow to better scrutinize whether the different data sets fulfill the prediction.

c) Some plots have very similar colors and are therefore hard to distinguish, e.g. the quadratic theory curve vs the basal data in Figure 2f. Also, in Figures 4f and 8e, the polygon type (number of neighbors n) is essential, but the colors in particular for n>5 are very hard to distinguish.

– Suggestions regarding point 4 of the public review (merely ideas):

a) As pointed out by Gómez-Gálvez et al., the curvature-based mechanism to create T1L transitions is due to a change in the aspect ratio of the 2D tissue cross section as one goes from apical to basal in an e.g. cylindrical tissue. One way to estimate the T1L rate that results from this might also be to use existing formalisms to understand how many T1 transitions are needed to create a certain change in 2D tissue aspect ratio (e.g. the texture tensor formalism, Graner et al., EPJE, 2008 and Guirao et al., *eLife*, 2015, or a triangle formalism, Merkel et al., PRE, 2017).

b) An interesting question could be how oriented the T1L transitions are ("How many of them do (net) contribute to the apico-basal change in 2D tissue aspect ratio?"), which one could address using orientational statistics of the T1 transitions (i.e. how uniform is the T1L orientation angle in the unfolded cylinder mantle).

c) Just a remark: I am happy to be convinced otherwise, but I would be surprised if one could indeed derive a dependence of the T1L transition number on cell neighbor number solely on geometric grounds (i.e. disregarding mechanics etc). This is because the curvature-based mechanism is about T1L transitions creating anisotropic deformation in the 2D cross section, while the cell neighbor number is a scalar (i.e. an "isotropic" quantity). I could be wrong here, but I would presume that any correlation between the two will depend on model details.

---

## [Author Response]

Essential revisions:1) Claims that imply causation but are based on correlations need to be either substantiated further to justify causation or toned down. Reporting only correlations might not suffice for publication in eLife.

As we emphasize also in the title of this paper, we can explain the observed 3D cell neighbour relationships with a minimisation of the lateral cell-cell surface contact energy. It is important to distinguish this from questions regarding the 3D cell shape, which we do not address. The confusion easily arises because our theory explains the polygon type and thus the shape of cross-sectional areas, but not their size. The size of the different cross-sectional areas, however, defines the overall 3D shape of the cell.

We show that, where present, the nuclear cross-sectional area is only slightly smaller than that of the cell, and the two measures correlate strongly (r = 0.94, Figure 5g). Referees 1 and 3 comment that this does not imply that the nucleus affects the cross-sectional area of the cell. We beg to differ here. The measured nuclear volumes are too large to allow a spherical nucleus to fit into a cylindrical cell of the measured height (as we now show explicitly in the new Figure 5h). Accordingly, to fit into the cell, the nucleus has to deform. Nuclei respond to external forces with anisotropic shape changes (Haase et al., 2016; Neelam et al., 2016), which is consistent with the elliptical nuclear shapes that we observe (Figure 5d). However, there is a limit to how much the stiff nucleus can deform (Lammerding, 2011; Shah et al., 2021), necessarily resulting in a local widening of the cell where the nucleus is present. Cell sections without nucleus typically have smaller cross-sectional areas, leading to a higher frequency of small cross-sections in cells compared to nuclei. We have added these additional explanations and references to the manuscript to strengthen the argument.

Nuclei in pseudostratified epithelia are well known to move continuously during the cell cycle, a phenomenon referred to as interkinetic nuclear migration (IKNM). The moving nuclei will continuously change the cross-sectional areas. In the live microscopy, the nuclei are not visible as we lack a live reporter for the nuclei. But given the strong correlation between the cross-sectional areas between nucleus and cell along the entire apical-basal axis and independent of the nuclear position and thus cell cycle phase (Figure 5g), and given the measured cross-sectional size of the nuclei, we believe that it is a safe inference that the nuclei are located at the wide parts of the cells, and these wide parts move during the cell cycle (Figure 8), consistent with IKNM.

In summary, we defined the physical principle behind the 3D cell neighbour relationships, and our data strongly suggest that the shape and movement of the nucleus is a key driver of the cell shape changes that translate into neighbour changes, both along the apical-basal axis and over time. Future work is required to unravel the physical determinants of the 3D shape of the epithelial cells and their nuclei.

2) Solve the issues spotted by Reviewer #3 in the theory presented in the SM (reviewer's point 4).

We would like to thank the referee for their careful checking of our theory. The comments helped us clarify the theoretical derivations. In the revised version, all issues have been fixed (for details, see point-by-point response). As they concerned only the formal derivation, but not the final outcome as shown in Figure 3g, the theoretical result and its interpretation are unaffected by these changes.

3) Estimate uncertainties as requested by Reviewer #3, point 3.

We have added the requested estimates of uncertainty by adding error bars showing the standard error of the mean.

4) Respond to the recommendations for the authors.

We thank the referees for their thoughtful comments, which we address below.

Reviewer #1 (Recommendations for the authors):L 60: Eq. (2) is wrong and should read instead 5+8/n.

Thanks for spotting the typo – we have corrected this.

Figure 1: I did not understand, why the authors reproduce the previously published data in Figure 1e-k. Figure 1d does not seem to be referred to in the text.

We reproduced the data to provide an introduction and context for the reader. The previous studies in planar epithelial surfaces included many more cells than we can 3D segment in the curved mouse lung epithelia. In the former, the phenomenological laws are therefore much better visible. Also, in light of the different referees’ comments, we believe that introducing Aboav-Weaire’s and Lewis’ laws clearly at the beginning is important to make our rather involved work accessible to a broad audience. We would therefore like to keep Figure 1 as it is.

Figure 2f: It is hard to distinguish between linear and quadratic dependences.

We are plotting the linear Lewis’ law and the quadratic relationship in comparison to the data. Given the small number of cells, the relationships are not as perfectly matched as for the planar apical sheets that we show in Figure 1. To improve visibility, we have adjusted the colors for basal and apical data in 2f.

Figure 3e: For tips there seems a clear bias for apical transitions. Could you comment on this?

The effect alluded to by the reviewer is likely to be an indirect consequence of the small number of cells considered for this dataset. Although nuclei are distributed along the entire apical-basal axis, consistent with pseudostratification, a few dividing cells in the tip dataset were very large, round, and apically located. This preferentially introduces apical cell geometry heterogeneity and cell neighbour rearrangements as dividing cells make room in an already crowded apical domain. This effect is likely to be more pronounced in a small dataset like this one (N=59) as this trend was no longer visible when we considered a dataset with many more cells (tube N=140). In the tube dataset with more cells, the fraction of T1 transitions appears uniform, except for the apical-most, and basal-most slice.

Ll 165: I did not understand this part. Why do you fit ellipses and not circles? Why do you consider two subsequent T1L transitions/consecutive neighbor number changes and not one?

Epithelial tubes in the developing lung have mostly collapsed rather than open and perfectly round lumens [Conrad et al., Development dev194209, 2021]. Consequently, their cross-sectional shape is not well approximated by circles. Instead, we find that ellipses typically approximate both their apical and basal shapes very well, which is why we fitted ellipses and not circles. To make this clearer to the readership, we have added this information to the paragraph in question. We have also extended Figure 3f to showcase the elliptic shape of the cross sections.

The reason why we considered the apicobasal distance between two consecutive T1L transitions rather than analysing the locations of individual transitions is that the former approach allows us to study the effect a change of tissue curvature has on them. Considering only individual T1L transitions and the tissue curvature they occur at would not provide any insight into the differential effect, i.e., how the transitions are affected by how quickly the tissue curvature changes. Our theory that we developed in the supplementary material quantifies the fold-change of tissue curvature between two T1L transitions. To allow for a fair comparison between theory and data, we analysed the data in the same way.

L 211: This statement contradicts later statements that involve the nucleus. Also, here you really write about minimizing a line energy, whereas it should be a surface energy; the lateral and the apico-basal directions are not independent.

Thanks for highlighting this important point so that we can further clarify. As we emphasize also in the title of this paper, our theory explains the 3D cell neighbour relationships, NOT the 3D cell shape. To be even more precise: while our theory does not attempt to explain the cross-sectional areas of the cells along their apical-basal axis (and thus the cell shape), it does explain the polygonal shape of each cross-section, and this then determines the number of neighbours that a cell has. Differently put: other processes (including the effect of the nucleus) define the cross-sectional areas of the cells, and, given the area distribution, the minimisation of the lateral cell-cell contact surface energy results in the observed polygon distribution, and thus cell neighbour relationship. To make this clearer, we have now rephrased the statement to read:

“We conclude that the neighbour relationships of epithelial cells along the entire apical-basal axis can be explained with a minimisation of the lateral cell-cell contact surface energy, as previously revealed for the apical layer.”

L 233: What do you mean by "pseudo-stratification"? Where is it shown?

The term pseudostratified epithelia is well established in the scientific literature on the subject and refers to epithelia which comprise only a single layer of cells, but have their cell nuclei positioned in a manner suggestive of stratified epithelia. The developing proximal mouse lung epithelium is well known to be pseudostratified (e.g. Que et al., Development, 2007, https://doi.org/10.1242/dev.003855), and we confirm in our measurements that the nuclei are indeed found along the entire apical-basal axis (Figure 5c). We are now citing a review by Caren Norden for readers not familiar with pseudostratified epithelia (https://doi.org/10.1242/jcs.192997).

Ll 241+ Figure 5d: There seems to be only a very weak correlation between nuclear volume and ellipticity or position along the apico-basal axis.

We agree with the referee, and we make no claim toward strong correlation. All we say is that they vary along the apical-basal axis, but there is at most a weak correlation. The only exception is close to the apical surface, where nuclei tend to be larger and more spherical, and we confirm this with a one-sided, two-sample Welch t-test.

Ll 244 + Figure 5f,g: Since the nuclei are inside the cell, could it be otherwise? Also, the correlation between cell and nuclear volume seems to be rather elevated. Is this correct?

Being inside the cell, the cross-sectional area of nuclei can, of course, not be larger than that of the cell, but it could be (substantially) smaller. As such, a strong correlation between the nuclear cross-section and cell cross-section is not an obvious necessity. As indicated in the figure panel, we find a sample correlation coefficient of 0.79 between nuclear and cell volume, which suggests strong association, likely because the nuclear volume expands in parallel to the cell volume during the cell cycle. We are now stating this explicitly in the text:

“The nuclear volumes are on average about 50% smaller than the cell volumes, and largely correlate (r = 0.79, Figure 5f), likely reflecting parallel expansion during the cell cycle.”

Ll 246: But it could also be the other way around or some independent factor could determine both. I do not think that the consistency you notice has any explanatory power.

We have now expanded on this statement to provide more context:

“Where present, the nuclear cross-sectional areas are only slightly smaller than those of the entire cell, and the cross-sectional areas of the cell and the nucleus are strongly correlated (r = 0.94, Figure 5g). The strong correlation can be accounted for by the opposing actions of cells and nuclei in the columnar epithelium. The nuclear volumes are too large to allow for a spherical nucleus to fit into a cylindrical cell of the measured height (Figure 5h). Accordingly, to fit into the cell, the nucleus necessarily has to deform. Nuclei respond to external forces with anisotropic shape changes (Haase et al., 2016; Neelam et al., 2016), which is consistent with the elliptical nuclear shapes that we observe (Figure 5d). However, there is a limit to how much the stiff nucleus can deform (Lammerding, 2011; Shah et al., 2021), resulting in a local widening of the cell where the nucleus is present. Cell sections without nucleus typically have smaller cross-sectional areas, thereby leading to a higher frequency of small cross-sections in cells compared to nuclei.”

As such, the sharp transition in cross-sectional area at the limits of the nuclei, and the concomitant T1 transitions, reflect the impact of the nucleus in locally widening the epithelial cell. This is what we meant to say, but we appreciate that this statement warranted a better explanation.

Figure 6: Are the results based on one explant?

This is correct. The analysis of dynamic cellular organization on the apical and basal cell surfaces was based on a single tissue explant. Lamentably, obtaining suitable specimens for morphometric analysis remains very challenging. Not only is the culturing and timelapse light-sheet microscopy imaging of mammalian rudiments complex, but target lung buds (such as the one in Figure 6) frequently present a high degree of heterogeneity in their shape – and the apical surfaces come so close that the tube lumen appears closed in many cases [Conrad et al., Development dev194209, 2021]. Naturally, inaccessible geometries like these did not allow for the analysis of both the basal and apical domains through 2.5D segmentation techniques, and as such, limited the overall number of samples included.

L 317: "Where present, the stiff nucleus determines the cell cross-sectional area, as is apparent from the strong correlation between the cell cross-sectional and the nuclear cross-sectional areas" Correlations do not imply causation. I do not think that you have data to support the causal relationship.

We absolutely agree of course; correlation does not imply causation. However, as explained above, we believe that our various data taken together strongly indicate that anything but causation is unlikely (see also answer to next point).

L 320: Again, this seems to be speculation. Figure 8 does not show the nucleus – and even if: from your the data it is not clear that the motion of the nucleus causes changes in the neighbor relationships.

As explained above, the nucleus cannot deform as much as would be required to fit into a columnar cell shape with perfectly straight lateral sides. The presence of nuclei therefore necessarily widens the cells locally. Thus, during nuclear migration, the location where the cell is locally widened by the nuclei moves with the nuclei. This is evidenced by the strong correlation in Figure 5g independent of nuclear position and thus cell cycle phase, which would not be expected to be found if it were otherwise. This in combination with the observation that T1L transitions are found with overwhelming predominance in the vicinity of the nuclei’s ends (Figure 5j) leaves us with the conclusion that the distinctly most plausible explanation is that nuclear migration alters the cell area, which in turn alters cell connectivity.

Reviewer #2 (Recommendations for the authors):Page 4, line 73, "following, Aboav-Weaire's law": I suggest replacing this with "following Aboav-Weaire's law" (i.e. remove the comma).

Thanks for spotting this typo.

Page 4, lines 89-90, "As such, growth and cell division determine the variability of the apical areas" and lines 100-101, "The cross-sectional areas vary as a result of cell growth, division and interkinetic nuclear migration (IKNM)": Maybe I missed it, but can the authors discount any role(s) for 'active' cell intercalations and/or cell death here?

We did not mean to exclude any such processes and have rewritten the two sentences to read:

“As such, active processes such as growth, cell division, cell death and extrusion, cell intercalation and apical constriction determine the variability of the apical areas and thus determine apical organisation indirectly.”

“The cross-sectional areas vary as a result of active cell processes, including interkinetic nuclear migration (IKNM).”

Page 4, lines 97-98, "the variation of […] define the": I suggest replacing this with "the variation of […] axis defines the" (i.e. add an 's').

Thanks for spotting this typo.

Page 5, lines 112-113, "We detected less than half as many cells on the apical side": Just to be clear here, do the authors mean that they could only detect half as many cells on the apical side due to their small size and/or lower resolution, or that there are really half as many cells on the apical side and hence over half the cells don't extend fully from the basal side to the apical side?

As rightfully speculated by the reviewer, the difference in detections between layers stems from the small size of cells on the apical side coupled with resolution limitations. Our analyzed explants had highly curved spherical/tubular morphologies with cells that fully extended from the apical to the basal layer – as is the case in pseudostratified tissues. However, this extreme curvature difference between tissue layers meant that while basal domains were large and easy to segment, cell geometries in constricted apical domains were small and hard to detect given the resolution of our Z.1 light sheet microscope.

Page 5, lines 118-119, "the neighbour changes in spherical geometries cannot be explained with such an effect (Gómez-Gálvez et al., 2018)": For clarity, I suggest briefly emphasizing why not (I guess the point is that as one looks along the apical-basal axis in this case, one is just seeing a dilation, so prismatic cells do the job fine).

As the referee correctly suggests, spherical shapes can be achieved with prismatic cells because the curvature change is the same in all directions, i.e., there is only a dilation. We have added a brief explanation at this position in the manuscript. We also discuss this in more detail in the next section in the context of figure 3f,g, and in the supplementary material.

Page 5, lines 127-130, "Aboav-Weaire's law (Figure 2e) and Lewis' law (Figure 2f) hold not only for the apical, but also for the basal datasets […] the apical layers, which have a larger area variability than the basal layers (Figure 2c), follow the quadratic law (yellow line) rather than the linear Lewis' law (black line)": I may simply have misunderstood things here, but the first sentence seems to indicate that Lewis' law holds for the apical dataset, while the second sentence seems to indicate that the apical layers follow the quadratic law rather than Lewis' law, which seems a contradiction. If I have indeed just misunderstood things here, then perhaps the authors could slightly rephrase these sentences for clarity.

As we explain in the Introduction, the linear Lewis’ Law and the quadratic law result from the same constraint for different area variabilities; hence the difference between the apical and basal surface. But we accept that it is confusing the way it was written, and we have rephrased this.

Page 7, lines 232-233, "giving the tissue a pseudostratified configuration": I suppose strictly speaking this is true of pseudostratified epithelia but not ALL epithelia, so it might be worth changing "In epithelia" to "In many epithelia" on line 227, or similar.

Thanks for catching this. We have added “pseudostratified” to the text, and we have added a reference that summarises nuclear positioning in different cells and tissues.

Page 9 lines 335-336, "signalling factors that control cellular tension are known to affect cell height (Widmann and Dahmann, 2009)": For clarity, perhaps the authors could name these signalling factor(s) explicitly here?

We have now added examples to the text:

“The determinants of the epithelial thickness, i.e., the distance between the apical and basal surfaces are still unknown, but signalling downstream of Fibroblastic Growth Factor (FGF), Sonic Hedgehog (SHH), Bone Morphogenetic Protein (BMP)/ transforming growth factor-β (TGF-β) and WNT has been observed to affect cell height, presumably via an effect on cell tension and/or cell-cell adhesion (Gritli-Linde et al., 2002; Hirashima and Matsuda, 2021; Kadzik et al., 2014; Kondo and Hayashi, 2015; Widmann and Dahmann, 2009).”

Page 16, Figure 1, panel (b): The chosen colour scheme makes it very difficult for me to tell the difference between 5, 6, 7 and 8 sided cells, since they're all shades of green. If possible, it would be great if the authors could use more contrasting colours.

We thank the reviewer for helping add clarity to our figures. The color palette has been adapted to increase the contrast and visibility of different cell types.

Page 16, Figure 1, panel (e): Can the authors briefly explain in the legend what the various abbreviations mean (cNT, EYE, dPE, dWL, dWP, dMWP, dPW, TP, dMWL), or at least broadly which different tissues/species are considered?

Thank you for pointing this out. We have amended the legend for Figure 1 to elaborate on these abbreviations.

Page 19, Figure 3, panel (b): I found this figure difficult to understand partly due to the colour scheme but also because the authors are plotting two datasets on the same image. If there is a way of simplifying this figure at all, that would be great.

We thank the reviewer for helping add clarity to our figures. The color palette has been adapted to increase the contrast and visibility of different cell types.

Reviewer #3 (Recommendations for the authors):– Citation of Rupprecht et al., MBoC, 2017 appears justified, as it is the first / one of the first papers to discuss apico-basal T1 transitions in epithelia based on experimental data.

We are now citing Rupprecht et al., MBoC, 2017 in the revised version. We apologise to the authors for the oversight of not including this paper in the original version.

– Recommendations regarding point 3. in the public review:a) Estimations of uncertainties (e.g. standard error of the mean) would be useful, e.g. in Figure 2 e,f. For instance, in Figure 2f, an indication for the stochastic uncertainty due to the finite sample size would be useful.

We have added error bars showing the SEM to all data shown in Figure 2f. In Figure 2e, we omitted the error bars because they are smaller than the symbols.

b) Aboav-Weaire law: For instance in Figure 2e, everything is close to a diagonal line. However, that there is a line with slope of approximately 6 in the plots of m_n*n over n is already a consequence of the Euler characteristic. To better display the correlation that is the core of the Aboav-Weaire law, I would suggest to directly plot m_n over n (as also one e.g. in Aboav, 1970). Together with error bars indicating uncertainty, this would allow to better scrutinize whether the different data sets fulfill the prediction.

Aboav-Weaire’s law is phenomenological and holds only approximately – the constants 5 for the slope and 8 for the intercept in Aboav’s original relationship are merely nearest-integer numbers. Deviations of the actual data from the line are therefore expected. We have described the deviation from the original simple straight line in much detail in an earlier publication [Vetter et al., bioRxiv 2019]. In Figure 2e, we plot the phenomenological relationship only for reference; the data is not expected to approach it with shrinking uncertainty bounds. Nevertheless, we agree that error estimates help judge the statistical significance with which we find the data to deviate from the simple phenomenological laws. In Figure 2e, the error bars are smaller than the symbols. But we have added error bars (SEM) to all data points in Figure 2f.

c) Some plots have very similar colors and are therefore hard to distinguish, e.g. the quadratic theory curve vs the basal data in Figure 2f. Also, in Figures 4f and 8e, the polygon type (number of neighbors n) is essential, but the colors in particular for n>5 are very hard to distinguish.

We thank the reviewer for helping add clarity to our figures. The color palette has been adapted to increase the contrast and visibility of different cell types.

– Suggestions regarding point 4 of the public review (merely ideas):a) As pointed out by Gómez-Gálvez et al., the curvature-based mechanism to create T1L transitions is due to a change in the aspect ratio of the 2D tissue cross section as one goes from apical to basal in an e.g. cylindrical tissue. One way to estimate the T1L rate that results from this might also be to use existing formalisms to understand how many T1 transitions are needed to create a certain change in 2D tissue aspect ratio (e.g. the texture tensor formalism, Graner et al., EPJE, 2008 and Guirao et al., eLife, 2015, or a triangle formalism, Merkel et al., PRE, 2017).

Thank you for these interesting references. Indeed, it appears that these other theoretical frameworks could offer a means of deriving similar (potentially equivalent) estimations of curvature effects on T1 transitions. As we were able to resolve all issues raised regarding our theory elsewise, we did not look into possible analogies here. We shall keep these references in mind for potential future considerations related to this aspect.

b) An interesting question could be how oriented the T1L transitions are ("How many of them do (net) contribute to the apico-basal change in 2D tissue aspect ratio?"), which one could address using orientational statistics of the T1 transitions (i.e. how uniform is the T1L orientation angle in the unfolded cylinder mantle).

We agree that this is an interesting aspect that could be worth investigating; thank you for this idea. A potential difficulty in interpreting the data might be that stress relaxation (visco-elastic and/or plastic) on the cellular level might affect the distribution of T1L orientations in a fashion that could be difficult to quantify. Nevertheless, this is an interesting future avenue that might be relatable to the cell orientation bias in the developing lung epithelium [Conrad et al., Development 148 (2021) dev194209].

c) Just a remark: I am happy to be convinced otherwise, but I would be surprised if one could indeed derive a dependence of the T1L transition number on cell neighbor number solely on geometric grounds (i.e. disregarding mechanics etc). This is because the curvature-based mechanism is about T1L transitions creating anisotropic deformation in the 2D cross section, while the cell neighbor number is a scalar (i.e. an "isotropic" quantity). I could be wrong here, but I would presume that any correlation between the two will depend on model details.

Thanks for the remark – we absolutely agree. Indeed, our data confirms this to some degree, in that we find that the observed number of T1L transitions does not agree with the geometrical curvature effect (Figure 3g), but is rather correlated with nuclear arrangement (Figure 5j), which in turn is undoubtedly strongly affected by various mechanical properties.